# Engineering rules that minimize germline silencing of transgenes in simple extrachromosomal arrays in *C. elegans*

Mohammed D. Aljohani [1,2], Sonia El Mouridi[1,2], Monika Priyadarshini[1,2], Amhed M. Vargas-Velazquez [1,2] & Christian Frøkjær-Jensen [1✉]

Transgenes are prone to progressive silencing due to their structure, copy number, and genomic location. In *C. elegans*, repressive mechanisms are particularly strong in the germline with almost fully penetrant transgene silencing in simple extrachromosomal arrays and frequent silencing of single-copy transgene insertions. A class of non-coding DNA, Periodic $A_n$/$T_n$ Clusters (PATCs) can prevent transgene-silencing in repressive chromatin or from small interfering RNAs (piRNAs). Here, we describe design rules (codon-optimization, intron and PATC inclusion, elevated temperature (25 °C), and vector backbone removal) for efficient germline expression from arrays in wildtype animals. We generate web-based tools to analyze PATCs and reagents for the convenient assembly of PATC-rich transgenes. An extensive collection of silencing resistant fluorescent proteins (e.g., *gfp*, *mCherry*, and *tagBFP*) can be used for dissecting germline regulatory elements and a set of enhanced enzymes (Mos1 transposase, Cas9, Cre, and Flp recombinases) enable efficient genetic engineering in *C. elegans*.

[1] King Abdullah University of Science and Technology (KAUST), Biological and Environmental Science and Engineering Division (BESE), KAUST Environmental Epigenetics Program (KEEP), Thuwal 23955-6900, Saudi Arabia. [2]These authors contributed equally: Mohammed D. Aljohani, Sonia El Mouridi, Monika Priyadarshini, Amhed M. Vargas-Velazquez. ✉email: cfjensen@kaust.edu.sa

Cells protect themselves by limiting foreign DNA expression, including transposons and transgenes, via small-RNA pathways and heterochromatin formation[1]. In mammals, conventional plasmid vectors are transcriptionally silenced in vivo in somatic cells, limiting our ability to develop efficient gene therapies[2]. Removing the bacterial backbone by recombination (minicircles)[3] or optimizing the codon-usage of transgenes[4] can increase expression. However, the effects of codon-optimization vary and may negatively impact the safety and efficacy of therapeutic proteins[5]. Alternatively, increasing the vector backbone's $A_n/T_n$ composition can reduce transgene-silencing for improved gene therapy[6]. Transgene silencing is not limited to animal cells. In plants, transgene silencing is also a significant roadblock to introducing desirable traits[7]. Transgene silencing often occurs in a stochastic manner over time, but expression can be stimulated through unknown mechanisms by the inclusion of introns[8]. In plants, silencing is linked to transgene structure, copy number, and double-strand RNA (dsRNA) generation[9]. Thus, despite some advances in limiting transgene silencing, this phenomenon remains a significant barrier to the development of transgenic technologies for biomedical and biotechnological use.

In the nematode Caenorhabditis elegans, the easiest and most commonly used method to generate transgenic animals is the injection of DNA into the germline syncytium, where semi-stable extrachromosomal and repetitive arrays are formed[10,11]. Persistent transgene expression in somatic cells is readily achieved from simple arrays[12], whereas germline expression is only observed for a few generations before transgene-silencing occurs[13]. This difference between cell types has likely evolved because the silencing of foreign DNA, such as transposable elements, is of particular importance in germ cells to prevent heritable defects leading to reduced fitness. Thus, germ cells face an inherent problem of balancing opposing pathways that repress and promote germline expression, respectively[14]. Candidate gene approaches[15–17], and genome-wide RNA interference screens[10,18] have identified small-RNA, chromatin, and splicing pathways that mediate active silencing processes in the germline. These studies have been essential for understanding the mechanistic basis of transgene silencing. However, they are of limited experimental use for biomedical or biotechnological transgene expression because mutant backgrounds frequently show germline defects that include maternal-effect sterility[19], accumulation of mutations caused by transposons[20], or progressive transgenerational loss of germline immortality[21].

Instead, several technical approaches independent of genetic background have been developed to overcome germline silencing of C. elegans transgenes: mimicking a genomic environment by co-injecting genomic DNA[13], low-copy transgene insertion by biolistic transformation[22], single-copy transgene insertion into defined or random genomic locations[23,24], and CRISPR/Cas9 insertion[25]. However, none of these methods entirely prevent silencing, and all are substantially more labor- and time-intensive compared with generating simple, extrachromosomal arrays[10]. From a scientific and practical perspective, there remains, therefore, considerable interest in understanding transgene silencing and developing methods that prevent this silencing.

A pervasive non-coding DNA structure named Periodic $A_n/T_n$ Clusters (PATCs) comprises a substantial fraction (6–10%) of the C. elegans genome[26]. We and others have previously shown that the inclusion of PATC-rich introns into single-copy integrated transgenes offered significant protection from germline silencing[27–31]. Together these experiments suggested that PATCs may be generally useful for protecting transgenes from silencing in the germline. However, the application of PATCs to prevent silencing has been limited by the technical difficulty of identifying PATC-rich DNA sequences and generating PATC-rich transgenes. Furthermore, best practices for generating silencing-resistant transgenes are spread over a disparate collection of published and unpublished manuscripts[32].

In this work, we develop a suite of tools and determine a set of engineering rules that largely prevent transgene silencing in simple extrachromosomal arrays in the germline of wildtype C. elegans. We develop an integrated web interface (www.wormbuilder.org/PATC/) that computes PATC scores and allows interactive browsing of pre-computed PATC values for all protein-coding genes in C. elegans. We show that insertion of PATC-rich introns in a one-pot reaction using validated and standardized reagents generates silencing-resistant transgenes. We use this ease of engineering transgenes and generating array animals to test the effects of codon adaptation, number and placement of introns, transgene concentration, optimal placement of fluorescent tags, temperature, and removal of the plasmid backbone, in addition to characterizing the protective effects of PATC-rich introns. Finally, we use these rules to generate collections of silencing-resistant fluorescent proteins that recapitulate endogenous gene regulation in the germline and high-efficiency gene-editing enzymes (e.g., Mos1 transposase and Cas9). In aggregate, these resources will broadly facilitate experiments across C. elegans laboratories.

## Results

**Online tools to identify and analyze PATC-rich sequences**. We developed a user-friendly, versatile web server to identify and quantify PATCs. The published PATC algorithm[26] is written in a narrowly used language (Pascal) and needs to be compiled for a specific operating system. Thus, the "activation energy" for studying PATCs is relatively high and requires a certain level of bioinformatics expertize. To facilitate the identification of PATCs, we updated our analysis of PATCs using the C. elegans genome build (ce11) and developed a set of online tools with an interactive graphical interface that can be accessed at www.wormbuilder.org/PATC/. The online app allows the computation of PATC content and phasing of any DNA sequence by either uploading a FASTA-formatted text file or by simple "copy-paste" (Fig. 1a). The tools also allow users to identify protein-coding genes with high PATC content or identify genomic regions with PATCs using a genome browser (Fig. S1). These tools make it significantly easier for other researchers to use or study the role of PATCs.

**Efficient generation of PATC-rich transgenes**. Our second aim was to facilitate the insertion of PATC-rich introns into transgenes for use in C. elegans or other organisms[6]. Incorporating introns is technically challenging with homology-based methods such as "Gibson" cloning[33]. Therefore, we developed an efficient protocol to insert up to four introns into a synthetic fluorescent proteins by Golden Gate cloning[34] (Fig. 1b) using a collection of PATC-rich introns (Table 1). Donor plasmids with introns do not contain splice acceptor and donor sequences, and are, therefore, compatible with species specific-splicing signals[35] incorporated in the synthetic "acceptor" transgene.

**PATC-rich transgenes are expressed in the germline from simple arrays**. An experimentalist building a transgene faces many design decisions: should the coding sequence be optimized? If introns are added, how many are necessary, and where should they be placed? Should these introns contain PATCs to mitigate silencing? Moreover, what regulatory elements (e.g., promoters and 3′-UTRs) and strategies for co-expression (e.g., operons or viral 2A peptides) are most efficient? What genetic contexts are compatible with expression? Is single-copy transgene integration

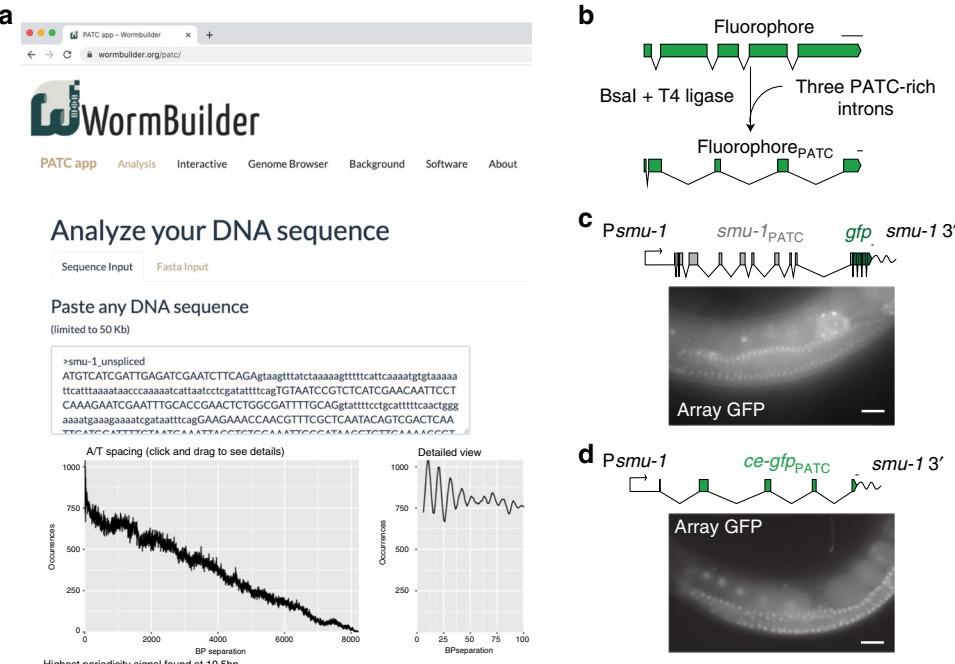

**Fig. 1 A set of tools to analyze and engineer transgenes with periodic A$_n$/T$_n$ clusters (PATCs). a** Screenshot of the graphical user interface from www.wormbuilder.org/PATC/. PATC values are calculated using the PATC algorithm described by Fire et al.[26] and a modified version ("balanced") described in Frøkjær-Jensen et al.[29]. The interactive and genome browser tabs allow visual inspection of pre-calculated genome-wide PATC values. **b** Synthetic introns in a transgene can be exchanged for PATC-rich introns by Golden Gate-mediated exchange using the type II BsaI restriction enzyme[34]. A library of PATC-rich donor introns (see Table 1) enables the routine exchange of three PATC-rich introns in a single reaction. **c** GFP expression in the germline of *C. elegans* from a simple extrachromosomal array carrying a P*smu-1*::*smu-1*::*gfp*::*smu-1* 3′-UTR transgene. Germline expression was independently verified in three independent biological replicates and in >30 independent transgenic lines. Scale bar = 20 µm. **d** Germline expression of a codon-optimized and PATC-rich *gfp* (*ce-gfp*$_{PATC}$) was expressed from simple extrachromosomal arrays using the *smu-1* promoter. Germline expression was independently verified in two independent biological replicates and in >25 independent transgenic lines. Scale bar = 20 µm. Transgene scale bars = 100 nucleotides.

**Table 1 Golden gate-compatible endogenous *C. elegans* introns.**

| Intron | Plasmid | Length (bp) | PATC density | Addgene |
|---|---|---|---|---|
| **Golden Gate Site 1** | | | | |
| *smu-1* | pCFJ1358 | 296 | 156 | #159804 |
| *smu-2* intron 3 | pCFJ1149 | 1176 | 605 | #161516 |
| ~250 bp | pCFJ2369 | 274 | 516 | #159877 |
| ~900 bp | pCFJ2214 | 900 | 1,250 | #159880 |
| Control (~250 bp) | pCFJ2365 | 276 | 10 | #159883 |
| **Golden Gate Site 2** | | | | |
| *smu-1* | pCFJ1359 | 313 | 98 | #159805 |
| ~250 bp | pCFJ2359 | 275 | 499 | #159878 |
| ~900 bp | pCFJ2259 | 967 | 896 | #159881 |
| Control (~250 bp) | pCFJ2345 | 277 | 15 | #159884 |
| **Golden Gate Site 3** | | | | |
| *smu-1* | pCFJ1360 | 966 | 98 | #159806 |
| ~250 bp | pCFJ2370 | 278 | 365 | #159879 |
| ~900 bp | pCFJ2215 | 935 | 1,210 | #159882 |
| Control (~250 bp) | pCFJ2366 | 265 | 17 | #159885 |

The PATC-score was calculated according to Fire et al. (2006).

into a safe harbor required, or will simpler extrachromosomal arrays suffice? If arrays are suitable, how much DNA needs to be injected, and what form of DNA (circular or linear) gives the most consistent expression? We set out to answer these questions and determine a set of practical and reliable design rules that minimize transgene silencing in the *C. elegans* germline from simple extrachromosomal repetitive arrays ("arrays"). We have focused on the germline as this tissue is the most difficult for

expression and is the subject of persistent efforts to understand small-RNA-mediated silencing mechanisms.

We based our initial transgene designs on two *gfp*-tagged genes, *smu-1* and *smu-2*, which are unusual because they are expressed in the germline from X-ray integrated simple arrays[36,37]. These genes are highly enriched for PATCs[26], suggesting that PATCs might generally enable germline expression from arrays. To determine germline expression requirements, we generated a *gfp*-tagged

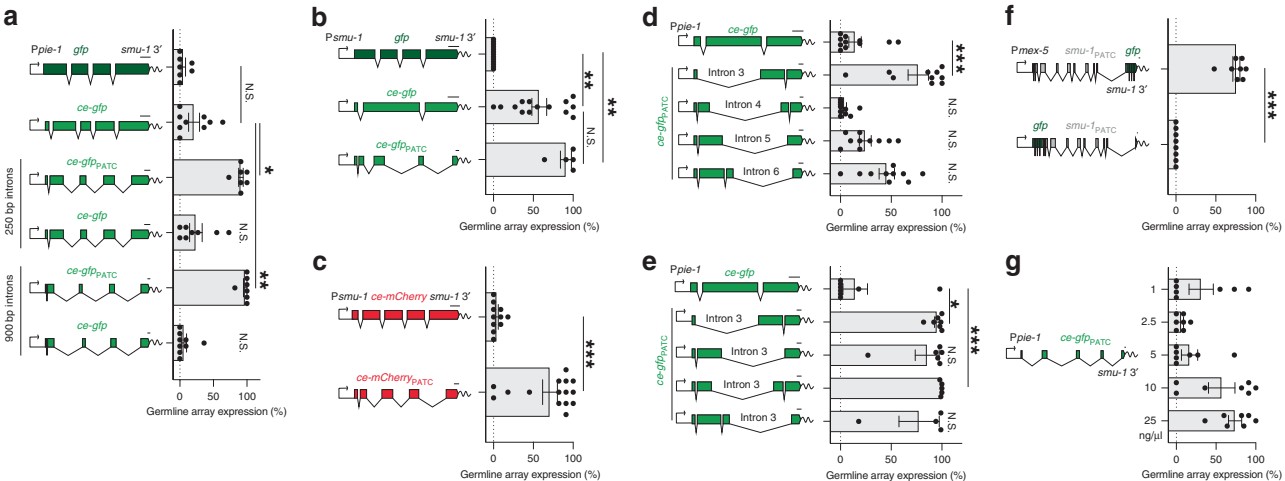

**Fig. 2 PATCs promote germline expression from simple extrachromosomal arrays. a** Quantification of germline GFP fluorescence from arrays carrying P*pie-1*::fluorescent protein::*smu-1* 3′-UTR transgenes. The *ce-gfp* transgenes contain synthetic introns (top) or substitutions with 250 bp and 900 bp endogenous introns with or without PATCs. $n = 7, 9, 7, 8, 7$, and 9 biologically independent transgenic lines (from top to bottom). **b** Germline GFP fluorescence from arrays with P*smu-1*::*gfp* transgenes. $n = 13, 14$, and 5 biologically independent transgenic lines (from top to bottom). **c** Germline array *mCherry* fluorescence from a codon-optimized *ce-mCherry* transgene[71] under control of P*smu-1*. $n = 11$, and 15 biologically independent transgenic lines (from top to bottom). **d** Germline array fluorescence of P*pie-1*::*ce-gfp* transgenes with single *smu-2* introns. $n = 11, 10, 10, 10$, and 10 biologically independent transgenic lines (from top to bottom). **e** Germline array fluorescence of P*pie-1*::ce-*gfp* transgenes with intron three from *smu-2* at various positions. $n = 8, 7, 5, 6$, and 4 biologically independent transgenic lines (from top to bottom). **f** Germline array fluorescence of an N- or C-terminal *gfp*-tagged *smu-1* transgene driven by the P*mex-5* promoter. $n = 8$ biologically independent transgenic lines for both conditions. **g** Germline fluorescence as a function of P*pie-1*::ce-*gfp* transgene concentration in simple extrachromosomal arrays (total concentration 100 ng/ul). $n = 7$ biologically independent transgenic lines for all conditions. All germline fluorescence was quantified from transgenic animals carrying simple extrachromosomal arrays imaged with a ×40 or ×63 oil objective at 25 °C. "*gfp*" (dark green) refers to the S65C GFP distributed in Fire lab vector kits (A. Fire, unpublished reagents). *ce-gfp* (light green) refers to a codon-optimized *gfp*[44] with piRNA homology removed[45]. PATC-rich transgenes are indicated with a subscript "PATC." Transgene scale bars = 100 nucleotides. Each datapoint indicates one independent measurement of germline fluorescence scored from 11 animals from an independent transgenic line. Bars indicate the mean, and error bars indicate the SEM. Statistics: **a–e** Kruskal–Wallis one-way ANOVA. Multiple comparisons: Dunnett's test. **c**, **f** Mann–Whitney two-tailed non-parametric test. $*p < 0.05$, $**p < 0.01$, $***p < 0.001$. Source data are available in the Source Data file.

*smu-1* transgene based on Spike et al.[36]. We observed reproducible germline (and somatic) expression from extrachromosomal arrays containing the *smu-1*::*gfp* transgene (Fig. 1c and Fig. S2). Arrays containing the *smu-1*::*gfp* transgene were expressed at high frequency using different combinations of germline promoters (P*smu-2*, P*pie-1*, and P*mex-5*) and 3′-UTRs (*smu-2* and *tbb-2*)[38,39] (Fig. S3). We conclude that genomic integration is not required for germline expression of a multi-copy array and that several commonly used promoters and 3′-UTRs can be used for expression.

These data suggest that sequences or signals intrinsic to the *smu-1*-coding region are significant determinants of germline expression from arrays. To define general rules for anti-silencing by PATCs, we designed transgenes encoding green fluorescent protein (GFP) that do not contain any homology to endogenous coding sequences or known piRNAs, which can silence germline genes[40]. These synthetic genes also lacks homology to 22 G RNAs, which are thought to protect genes from silencing through a pathway dependent on the Argonaute protein CSR-1[41–43]. The transgenes were designed using a popular web-based platform for *C. elegans* codon adaptation[44], and a custom algorithm to eliminate sequences homologous to known piRNA sequences[40,45]. We generated several GFP variants that contained several (two to four) small, synthetic introns that were designed to enable subsequent insertion of large stretches of PATC-rich sequence. When this optimized "*ce-gfp*PATC" containing PATC-rich introns was inserted between the promoter and 3′-UTR from the *smu-1* gene, GFP was robustly expressed (Fig. 1d).

We generated additional *ce-gfp* transgenes using our Golden-Gate-based cloning approach (Fig. 1b) to test the role of PATC-

rich introns in isolation. We exchanged the synthetic introns in *ce-gfp* for 250 bp and 900 bp native introns with or without PATCs and quantified germline expression from arrays using a *pie-1* promoter (Fig. 2a). A commonly used *gfp* that is not codon-optimized for *C. elegans* (distributed in the Fire lab vector kits, Andrew Fire unpublished reagents) was expressed at low frequency in the germline. Codon-optimization modestly increased the frequency of expression but did not reach statistical significance (Fig. 2a). In contrast, germline expression was observed at high frequency in all animals with arrays carrying *ce-gfp*PATC transgenes with native 250 bp and 900 bp introns. To address whether the enhanced germline expression was the result of using native introns, we replaced the PATC-rich introns with 250 bp and 900 bp endogenous introns lacking PATCs and observed no enhanced germline expression (Fig. 2a).

We tested the modest effect of codon-optimization using the *smu-1* promoter (Fig. 2b). With P*smu-1*, we observed substantial germline array expression of *ce-gfp* with synthetic introns and a further increase from adding PATC-rich introns (Fig. 2b). A codon-optimized *mCherry* (*ce-mCherry*) with synthetic introns was poorly expressed using P*smu-1*, but PATC-rich introns significantly increased expression (Fig. 2c). Similarly, a *smu-1* promoter from *C. briggsae* (P*cbr-smu-1*) also required addition of PATCs for robust germline expression (Fig. S4). These results demonstrate that codon-optimization in itself does not necessarily ensure germline expression.

How many PATC-rich introns are required for this anti-silencing effect? We tested the effect of individual *smu-2* introns (rather than all four *smu-2* introns as in Fig. 1d). A single intron (intron 3) significantly stimulated germline expression (Fig. 2d).

In contrast, two *smu-2* introns (introns 5 and 6) resulted in only a modest but non-significant increase in expression, and one intron (intron 4) had no effect (Fig. 2d). This unequal effect was not due to where the introns were positioned in the transgene as intron three increased germline expression from arrays at all four locations (Fig. 2e). Combining intron three with the other *smu-2* introns did not further increase expression, and artificial introns with high levels of PATCs were ineffective at increasing expression (Fig. S5). Although codon-optimization and inclusion of PATCs are generally effective improvements, we were unable to generate a silencing-resistant tandem dimer Tomato (*tdTomato*)[46]. We observed only infrequent expression in the germline that was prone to rapid bleaching (Fig. S6). *tdTomato* silencing was not due to the two tandem repeats as an analogous tandem dimer *ce-gfp* was expressed at high frequency (Fig. S6). Although not perfect, we conclude that PATC-rich introns generally stimulate germline expression from arrays and that not all PATC-rich introns are equally effective.

Fluorescently tagged endogenous genes inserted as single-copy transgenes by MosSCI are sensitive to germline silencing depending on where the *gfp* is positioned. For example, *rde-3* and *cdk-1* with N-terminal *gfps* were frequently silenced, whereas C-terminally tagged genes were rarely silenced[47]. We observed the same effect in arrays: a *smu-1::gfp* transgene was frequently expressed in the germline whereas a *gfp::smu-1* transgene was consistently silenced (Fig. 2f). Thus, we suggest inserting foreign DNA sequences at the C-terminus of endogenous genes, if possible, for optimal germline expression.

Complex extrachromosomal arrays approximate euchromatin by injecting high concentrations of genomic carrier DNA (50–100 ng/ul) and low concentrations of the transgene (1–2 ng/ul), which can prevent germline silencing[13]. Complex arrays are infrequently used as they are challenging to generate, and expression is not easily maintained. To test if our experimental conditions are similar to complex arrays, we tested the effect of transgene concentration on germline expression (Fig. 2g). For simple arrays, germline expression of P*pie-1*::*ce-gfp*$_{PATC}$ increased with higher transgene concentrations (Fig. 2g). Furthermore, PATC-rich carrier DNA did not prevent transgene silencing. Instead, high concentrations of carrier DNA with PATCs occasionally caused unusual aberrant germline morphology (Fig. S7). We, therefore, recommend injecting transgenes at high concentrations and using standard DNA ladder as carrier DNA.

In aggregate, these results establish a basic set of design rules that improve germline expression. Germline expression from transgenes in simple arrays is possible, and PATC-rich introns stimulate expression. We have found no evidence that PATC-rich promoters or 3′-UTRs improve germline expression. Transgenes consisting of fluorescently tagged endogenous genes are more efficiently expressed when tagged at the C-terminus, which is in agreement with previous observations from single-copy insertions[47]. Finally, higher transgene concentrations in the injection mix result in more-frequent germline expression but inclusion of PATC-rich carrier DNA appears to be toxic. Although we primarily focused on determining a set of applicable rules for improved germline expression, two of our results provide biological insight into requirements for transgene expression. First, there is no strict requirement for protein-coding sequences from endogenous genes (*smu-1* or *smu-2*), suggesting that PATCs and the *csr-1* (which depends on homology to coding regions) pathway are complementary. Second, array silencing in the germline was proposed to result from divergent transcription and dsRNA intermediates leading to RNAi-mediated silencing[48]. This model is difficult to reconcile with the observation that higher plasmid concentrations increase germline expression unless PATCs prevent antisense transcription.

## Early introns and backbone removal increase germline expression

Transposons are detected in the yeast *Cryptococcus Neoformans* by nuclear RNAi machinery that scans for sub-optimal introns[49]. In *C. elegans*, a similar mechanism has been proposed, with introns acting as a barrier to repressive nuclear RNAi pathways that act via the EMB-4 RNA helicase[15]. Perhaps, N-terminal *gfp*-tagged genes are prone to silencing owing to unusual codon-usage or intron structure, specifically at the 5′ end of genes? To explore this possibility, and the general requirement for introns to enable germline expression, we generated chimeric transgenes consisting of *smu-1* genomic DNA and cDNA with a C-terminal *gfp* tag (Fig. 3a). We observed robust germline fluorescence from transgenes with introns at the 5′ end of the gene but virtually no expression from a *smu-1* cDNA or a *smu-1* mini-gene lacking the first five introns (Fig. 3a). A single synthetic intron at the 5′ end of *smu-1* restored germline fluorescence, establishing that endogenous introns are not required. Similarly, trans-spliced promoters (that have "half" of a splicing reaction in the 5′ -UTR) also partially restored germline expression from arrays in the absence of 5′ introns (Fig. S8). This improved expression could be due to improved transcription and mRNA processing or by enhanced translation efficiency[50]. Surprisingly, the *smu-1* cDNA transgene was expressed in the germline from single-copy transgene insertions, whereas the chimeric transgene remained silenced, showing that transgene context can play a role in transgene silencing or detection (Fig. 3b).

We tested the role of splicing in detail by generating P*smu-1*::*ce-gfp* transgenes with a variable number of synthetic introns (all lacking PATCs) at various locations and monitoring germline expression from arrays (Fig. 3c–e). *ce-gfps* with no introns or a single intron were infrequently expressed in the germline (Figs. 3c and S9). In contrast, *ce-gfps* with two introns were expressed at consistently high frequency when one intron was located near the 5′ end of the coding region (at base number 48) (Fig. 3c). Further experiments showed that efficient germline expression required a short first exon (<350 base pairs and preferably shorter than 150 base pairs) combined with a second intron anywhere (Fig. 3d). This is similar to observations in human cells, where short first exons (~250–500 base pairs) serve as position-dependent transcriptional enhancers that act via activating histone modifications (H3K4me3 and H3K9ac) leading to higher expression levels[51]. Furthermore, short first exons promote transcriptional accuracy and reduce antisense transcription by repressing transcriptional initiation within the first exon[51]. Antisense transcription is a potent trigger for small-RNA mediated silencing in the germline[52] and transgenes with long first exons may, therefore, be actively silenced. In support of this model, Makeyeva and colleagues have recently shown that genes from which introns have been removed become default targets of small-RNA silencing in the germline (Y.V. Makeyeva and C.C. Mello, personal communication 2020).

Are transgenes with no introns at the 3′ end of the coding region expressed at all? We tested several *ce-gfp* transgenes in different genetic contexts to determine whether they were expressed in some circumstances (Figs. 3e and S9). We observed robust germline expression from these sub-optimal single-copy transgenes when inserted into a permissive genomic environment (*ttTi5605*), illustrating the importance of transgene copy number and chromatin context for germline silencing. Somatic transgene expression in *C. elegans* is improved by removing the plasmid backbone by PCR amplification or restriction digest and gel purification[53], similar to how backbone removal increases the perdurance of transgene expression in mammals[3]. We observed frequent germline expression when *ce-gfp* transgenes were PCR amplified or gel-purified (Fig. 3e). Restriction enzyme digestion of the vector backbone alone was also sufficient to increase germline

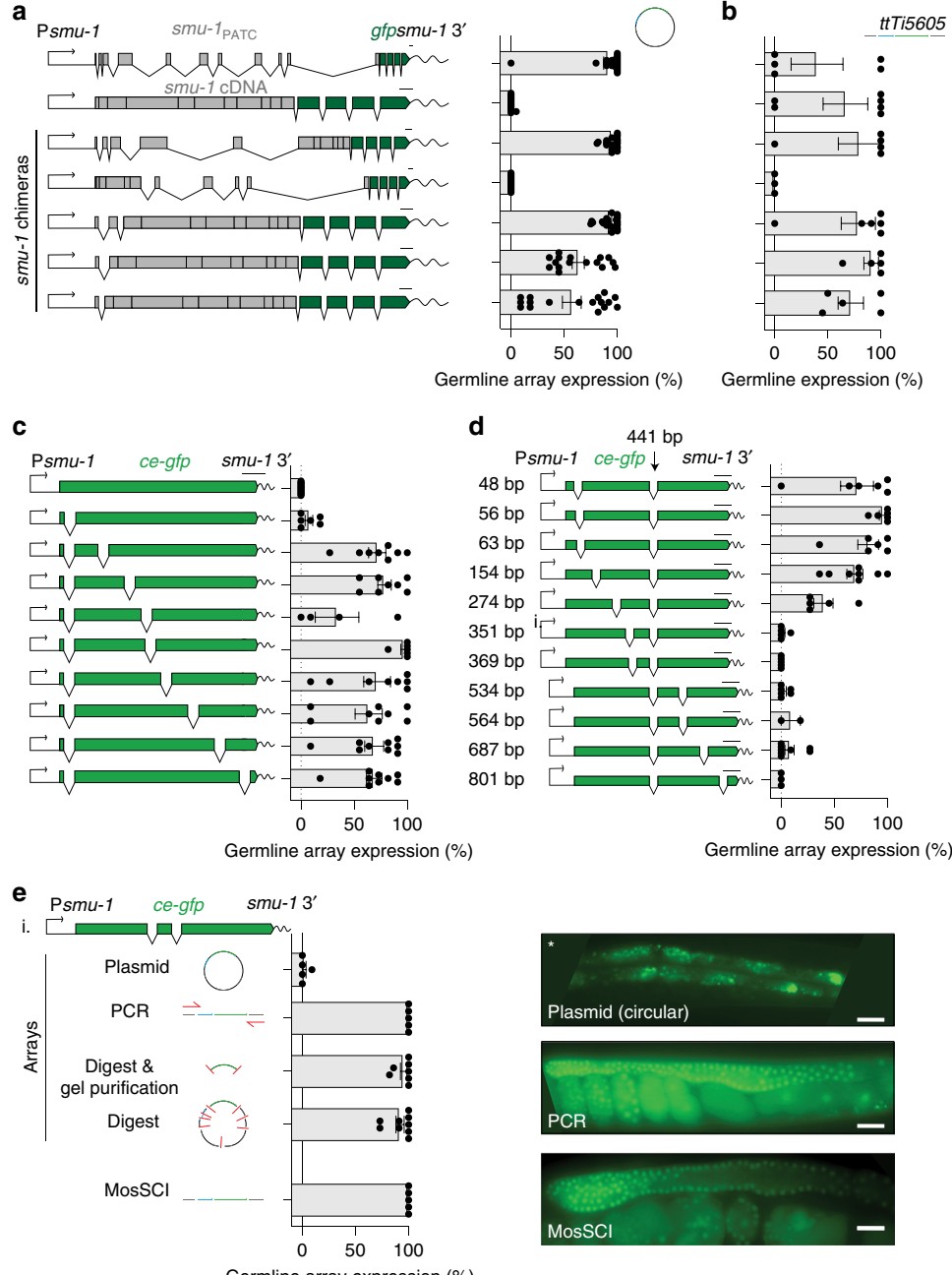

**Fig. 3 The effects of splicing and the plasmid backbone on germline expression. a** Germline expression of chimeric *gfp*-tagged *smu-1* cDNA and genomic transgenes from simple arrays. n = 22, 20, 16, 27, 16, 17, and 17 biologically independent transgenic lines (from top to bottom). **b** Germline expression of chimeric *gfp*-tagged *smu-1* cDNA and genomic transgenes from single-copy MosSCI insertions at the permissive *ttTi5605* insertion site (Chr. II). n = 5, 6, 5, 3, 6, 5, and 5 biologically independent transgenic lines (from top to bottom). **c** Simple array germline expression of *ce-gfp* containing no synthetic introns (top), one intron, or two introns expressed from a *smu-1* promoter. n = 15, 6, 8, 8, 4, 6, 8, 8, 9, and 9 biologically independent transgenic lines (from top to bottom). **d** Germline expression of *ce-gfp* containing two early or late synthetic introns expressed from a *smu-1* promoter from simple arrays. n = 6, 6, 6, 8, 5, 5, 5, 6, 2, 8, and 3 biologically independent transgenic lines (from top to bottom). **e** The effect of transgene context on expression of *ce-gfp* containing two synthetic introns using the *smu-1* promoter. Images (right) show typical GFP expression. n = 5, 5, 7, 9, and 5 biologically independent transgenic lines (from top to bottom). Scale bar = 20 μm. * = non-specific gut-granule fluorescence. Conditions: plasmid = simple arrays with circular plasmid at 25 ng/ul. MosSCI = single-copy transgene insertion at *ttTi5605* (Chr. II). PCR = simple arrays with PCR amplified transgene (no plasmid backbone). Digest & gel purification = KpnI, EcoRV, and ApaLI digest with the transgene coding region isolated by gel electrophoresis and gel purification (backbone removed). Digest = KpnI, EcoRV, ApaLI digest, and bulk purification over a column (backbone digested but not removed). Transgene scale bars = 100 nucleotides. All germline fluorescence was quantified from transgenic animals imaged with a ×40 or ×63 oil objective at 25 °C. Each datapoint indicates one independent measurement of germline fluorescence scored from 11 animals from an independent transgenic line. Bars indicate the mean, and error bars indicate the SEM. Source data are available in the Source Data file.

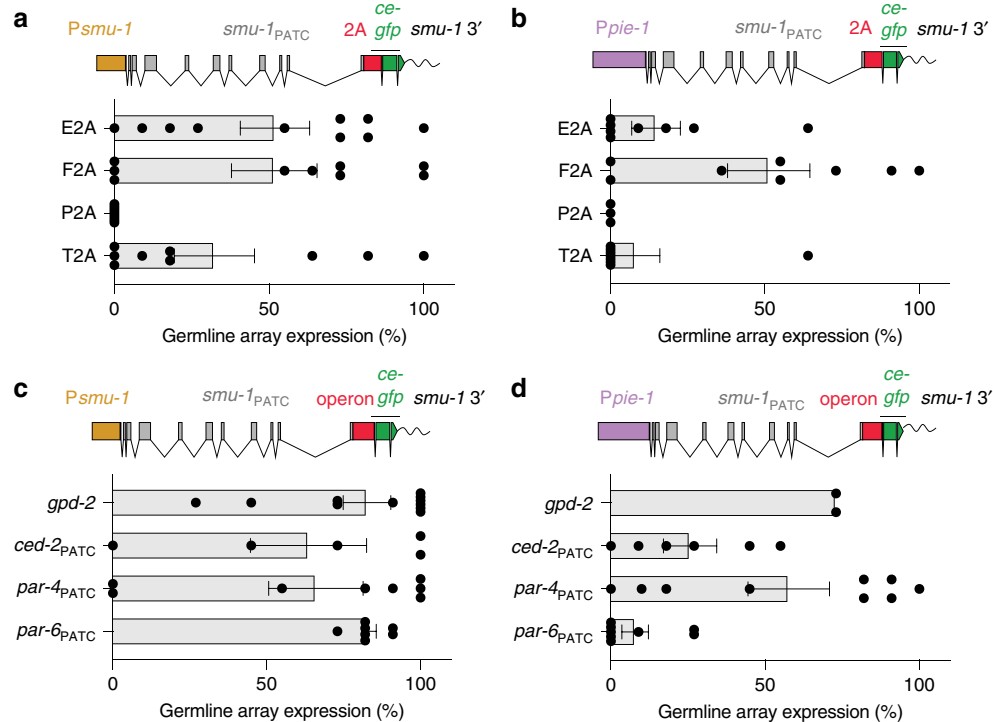

**Fig. 4 The effect of viral 2A peptides and operon sequences on germline co-expression of genes. a** Germline expression of a *ce-gfp* fused to a genomic P*smu-1*::*smu-1* gene (germline and soma) separated by various viral 2A peptide tags. 2A tags result in ribosomal skipping and peptide cleavage, which allows co-expression of two or more genes from one open reading frame[54]. *n* = 10, 9, 8, and 9 biologically independent transgenic lines (from top to bottom). **b** Germline expression of a *ce-gfp* fused to a genomic P*pie-1*::*smu-1* gene (germline-specific) separated by various 2 A peptide tags. *n* = 8, 8, 3, and 8 biologically independent transgenic lines (from top to bottom). **c** Germline expression of a *ce-gfp* fused to a genomic P*smu-1*::*smu-1* gene (germline and soma) separated by various operon sequences. Operons are common in *C. elegans*[55], particularly for genes expressed in the germline[56]. Operons allow co-expression or more than one gene from a single promoter. *n* = 11, 5, 8, and 7 biologically independent transgenic lines (from top to bottom). **d** Germline expression of a *ce-gfp* fused to a genomic P*pie-1*::*smu-1* gene (germline and soma) separated by various operon sequences. *n* = 2, 6, 9, and 8 biologically independent transgenic lines (from top to bottom). Transgene scale bars = 100 nucleotides. All germline fluorescence was quantified from transgenic animals carrying simple extrachromosomal arrays imaged with a ×40 or ×63 oil objective at 25 °C. Each datapoint indicates one independent measurement of germline fluorescence scored from 11 animals from an independent transgenic line. Bars indicate the mean, and error bars indicate the SEM. Source data are available in the Source Data file.

fluorescence (Fig. 3e), a somewhat more convenient approach for large transgenes. Many of our experiments were done using the pCFJ150 backbone for *cbr-unc-119*(+) selection in arrays and to make transgenes compatible with single-copy insertion. The pCFJ150 backbone contains two 1.5 kb genomic homology regions (in addition to the 2.1 kb *cbr-unc-119* selection marker) flanking the transgene, which could conceivably shield from silencing. However, transgenes inserted into a backbone vector (pDESTR4-R3) with no other nematode DNA were expressed in the germline at similar or higher frequency (Fig. S10). Other vector backbones may result in reduced or increased silencing from circular plasmids, but cloning is not limited to a single vector.

We conclude that the inclusion of two introns, with one in the first 150 base pairs of the coding sequence and removing the cloning vector backbone, stimulates germline expression. These observations raise questions about how the germline's silencing machinery identifies and silences foreign DNA elements based on a combination of copy number and transgene structure.

**Germline co-expression using viral 2A peptides and operons**. The relative ease of expressing a fluorescent protein located at the 3′-end of a PATC-rich gene suggested that it might be possible to bypass silencing by expressing transgenes downstream of an endogenous gene (e.g., a *gfp* downstream of *smu-1*). We tested this strategy using two different methods for co-expressing genes in *C. elegans*: viral 2A peptides and operons. 2A peptides allow

co-expression of two or more genes by ribosomal skip mechanisms that occur during protein translation, an approach that has been validated in *C. elegans*[54]. We tested four different 2A peptide sequences (E2A, F2A, P2A, and T2A) for expressing *ce-gfp* downstream of a full-length *smu-1* gene using *smu-1* or *pie-1* promoters (Fig. 4a, b). Three of the four 2A peptides allowed co-expression but at reduced frequency compared to *smu-1*::*gfp* fusions, despite codon-optimizing *ce-gfp* (Fig. 4a, b).

The second co-expression strategy relied on endogenous operons[55], which is a common organization for germline-expressed genes[56]. We tested *smu-1* and *ce-gfp* co-expression using the frequently used *mai-1*/*gpd-2* operon and three additional operons with high PATC content (Fig. 4c, d). All four operons allowed germline expression at high frequency from P*smu-1* (Fig. 4c), whereas *gpd-2* and *par-4* resulted in more-frequent expression when using a *pie-1* promoter (Fig. 4d). These results indicate that high PATC content in intergenic operon sequences is unnecessary and does not promote germline expression from arrays.

We conclude that a transgene (*ce-gfp*) can be expressed downstream of 2A peptides and intergenic operon sequences. Owing to the higher efficiency and native function in *C. elegans*, we recommend using the *gpd-2* operon sequence for this strategy.

**Germline expression is stable but temperature dependent**. In *C. elegans*, growth at high temperature (25 °C) partially prevents

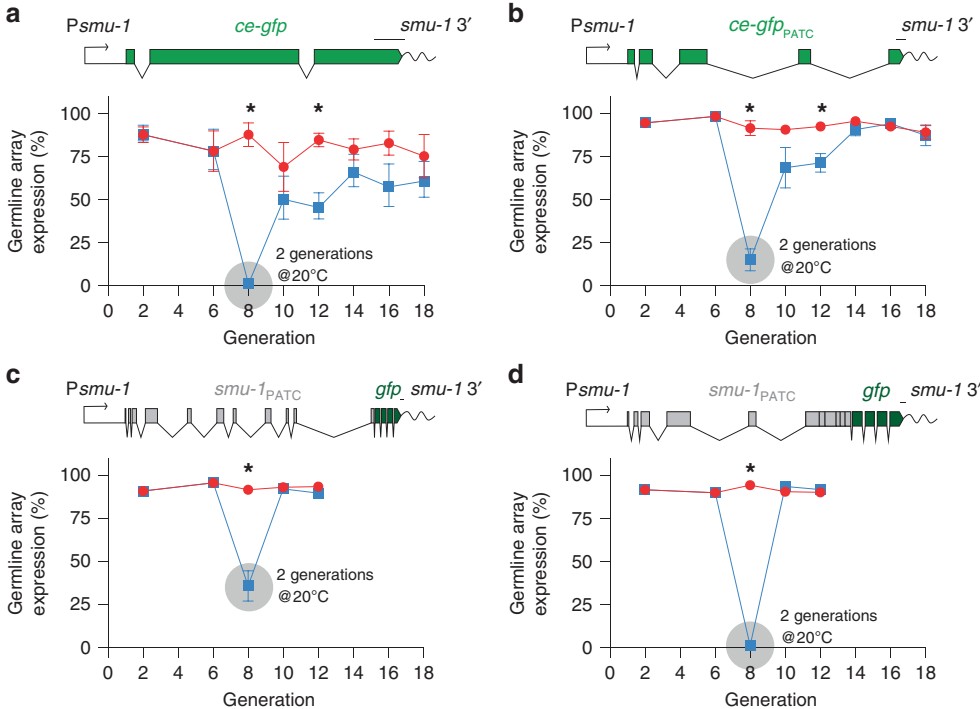

**Fig. 5 Reversible, temperature-dependent silencing of PATC-rich transgenes.** Quantification of GFP fluorescence from extrachromosomal arrays in the germline over several generations. Animals were grown for two generations at a lower temperature (20 °C) (blue line) compared with controls (red line) propagated continuously at 25 °C. **a** Germline expression of ce-GFP containing no PATCs under the PATC-rich smu-1 promoter and 3′-UTR. n = 8 biologically independent transgenic lines. **b** Germline expression of ce-GFP$_{PATC}$ under the PATC-rich smu-1 promoter and 3′ UTR. n = 11 biologically independent transgenic lines. **c** Germline expression of a gfp-tagged "full" length Psmu-1::smu-1 transgene. n = 11 biologically independent transgenic lines. **d** Germline expression of a gfp-tagged chimeric Psmu-1::smu-1 transgene with the last five introns removed. n = 11 biologically independent transgenic lines. Transgene scale bars = 100 nucleotides. All germline fluorescence was quantified from transgenic animals carrying simple extrachromosomal arrays imaged with a ×40 or ×63 oil objective at the indicated temperatures. Each datapoint indicates one independent measurement of germline fluorescence scored from 11 animals from an independent transgenic line. Bars indicate the mean, and error bars indicate the SEM. Statistics: Two-way ANOVA and repeated t test corrected for multiple comparisons with the Holm–Sidak method. *p < 0.05. Source data are available in the Source Data file.

gradual silencing of germline-expressed transgenes[57] via poorly understood mechanisms. All experiments described until now were therefore performed at 25 °C. We tested long-term germline expression and temperature-dependent silencing by establishing transgenic lines at 25 °C and transiently shifting one group of animals to 20 °C for two generations. For Psmu-1::ce-gfp transgenes, we observed persistently high expression for many generations at 25 °C but gradually reversible silencing at 20 °C (Fig. 5a, b). Transgenes containing endogenous coding sequences encoded by full-length and chimeric smu-1::gfp showed similar temperature-dependent silencing with one difference: smu-1 transgenes were fully de-silenced in the first generation after returning animals to 25 °C (Figs. 5c, d, and S11).

We conclude that transgenes in simple arrays can be indefinitely expressed in the germline when animals are maintained at 25 °C. The expression state can be reversed over a few generations by switching between 20 °C and 25 °C. To our knowledge, such reproducible and full reversibility has not been observed before and could be a useful paradigm for studying mechanisms that lead to transgenerational silencing in response to a simple environmental change[58].

**PATC-rich transgenes recapitulate endogenous germline expression.** Germline expression from PATC-rich transgenic arrays may facilitate experimentation to understand germline regulatory elements (e.g., promoter bashing[59] or 3′-UTR regulation[38]). However, such experiments depend on PATC-rich

introns not influencing expression themselves by, for example, acting as enhancers.

To test if more accurate promoter expression patterns can be captured from expressing optimized fluorescent proteins in arrays, we tested a 4.7 kb promoter from synaptobrevin (Psnb-1) driving the expression of three fluorescent proteins: gfp, ce-gfp, and ce-gfp$_{PATC}$. Synaptobrevin has a role in neurotransmission and is primarily expressed in neurons based on antibody staining[60]. However, mRNA expression suggests expression in the germline[61]. Transgenic animals with arrays showed consistent germline expression only when using the ce-gfp$_{PATC}$ transgene (Fig. 6a). Germline expression is unlikely to be driven by endogenous germline enhancers in the PATC-rich introns because the same PATC-rich ce-gfp was not expressed in the germline when paired with the minimal pes-10 promoter[62]. The absence of germline expression is not because enhancers placed downstream of the minimal promoter are not active; tissue-specific enhancers[63] inserted into PATC-rich introns yielded expression in seam cells and the ventral cord neurons (Fig. 6b). We note an important caveat to these experiments: germline and somatic promoters may have fundamentally different architectures[61]. Therefore, the minimal promoter from the "soma only" pes-10 gene[62] may not accurately capture germline enhancer activity. However, to our knowledge, no alternative minimal promoter has been used to study germline enhancers. PATC-rich fluorescent proteins could help identify germline-specific minimal promoters and experimental validation of differences between germline and somatic promoters.

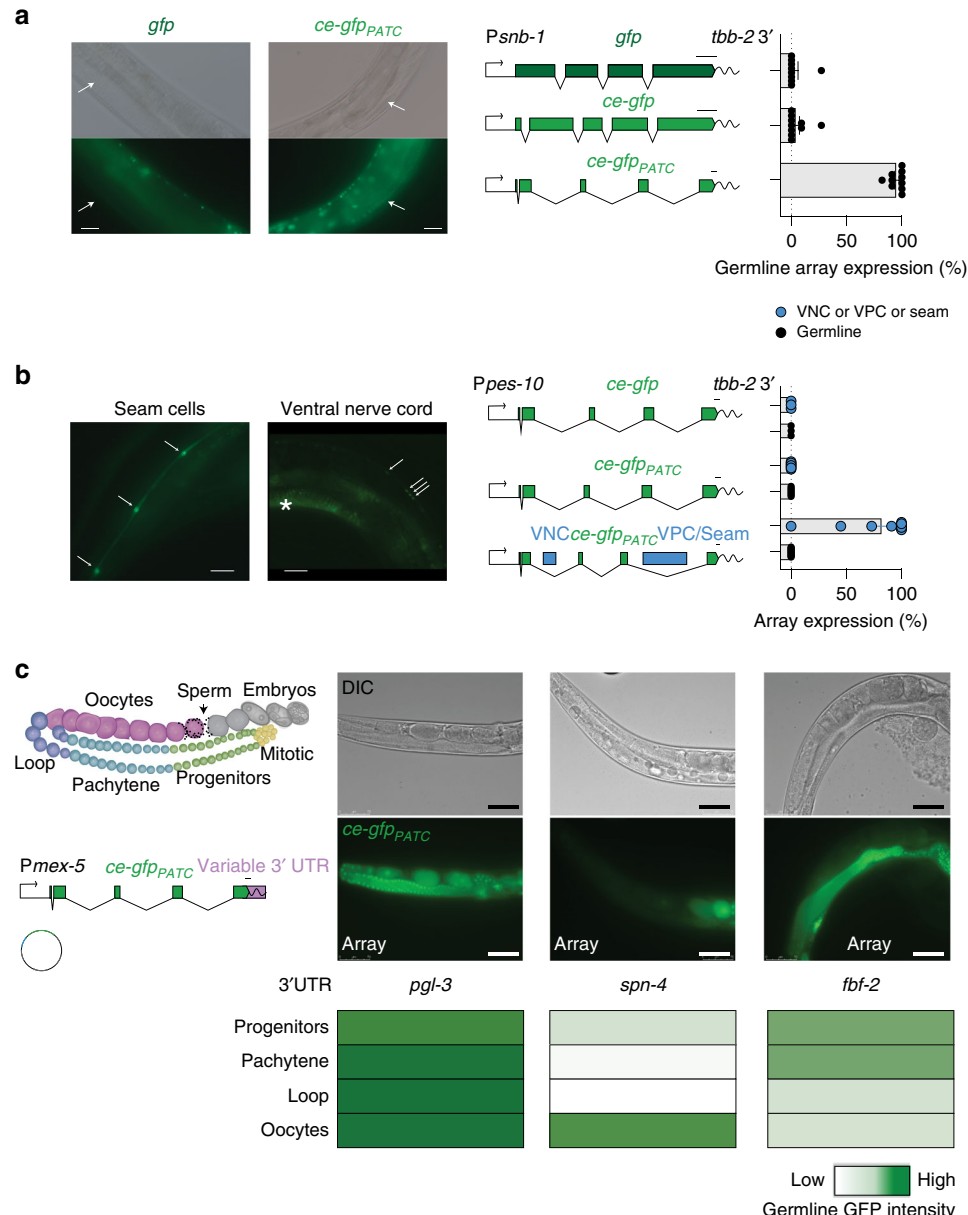

**Fig. 6 PATC-rich fluorescent proteins are not ectopically expressed. a** Images of GFP expression from simple extrachromosomal arrays with transgenes expressing *gfp* ("Fire lab GFP") and *ce-gfp* (codon-optimized GFP) under control of the synaptobrevin (*snb-1*) promoter. The white arrowhead indicates germ cells. *n* = 9, 10, and 10 biologically independent transgenic lines (from top to bottom). **b** Test of enhancer activity from PATCs. Images of GFP expression in animals from arrays carrying a minimal *pes-10* promoter fused to *ce-gfp*s with 900 bp introns with no PATCs (top), 900 bp introns with PATCs (middle), or 900 bp introns with PATCs and enhancers (blue rectangles) for the ventral cord neurons (VCNs) or seam cell + ventral precursor cells (VPCs)[63]. Images show GFP expression in seam cells (left) and ventral cord neurons (right) indicated by white arrows. * indicates the non-specific gut-granule fluorescence. *n* = 3, 7, and 11 biologically independent transgenic lines (from top to bottom). **c** 3′-UTR control of stage-specific germline expression. P*mex-5*::*ce-gfp*_PATC transgenes were expressed from arrays with several different 3′-UTRs that are known to mediate stage-specific germline expression[38]. The relative fluorescence intensities are indicated in heat-maps below representative images of transgenic animals. *n* = 6 biologically independent transgenic lines for all 3′-UTRs. Transgene scale bars = 100 nucleotides. *gfp* = GFP(S65C) from Fire lab vector kit (A. Fire, unpublished reagents). *ce-GFP* = codon-optimized GFP[44] with piRNA homology removed[45]. Fluorescence was quantified from transgenic animals carrying simple extrachromosomal arrays imaged with a ×40 or ×63 oil objective at 25 °C. Image scale bars = 50 microns. Each datapoint indicates one independent measurement of fluorescence scored from 11 animals from an independent transgenic line. Bars indicate the mean, and error bars indicate the SEM. Source data are available in the Source Data file.

3′-UTRs regulate stage-specific expression within the germline[38], and we tested if optimized fluorescent proteins capture this regulation. We generated P*mex-5*::*ce-gfp*_PATC transgenes with two 3′-UTRs known to regulate gene expression in specific regions of the germline (*fbf-2* and *spn-4* 3′-UTRs), and one 3′-UTR that permits ubiquitous expression in all germ cells (*pgl-3* 3′-UTR)[38].

All these constructs showed the expected expression patterns from arrays (Fig. 6c).

We conclude that arrays with *ce-gfp*_PATC transgenes can accurately report known germline regulation via promoters and 3′-UTRs. We have generated and validated an extensive collection of codon-optimized fluorescent proteins with or without PATCs

**Table 2 Optimized fluorescent proteins and tags.**

| Characteristics | | | Optimized fluorophore | | | Germline optimized (with PATCs) | |
|---|---|---|---|---|---|---|---|
| | CAI score | piRNA score | Cytosolic | Nuclear | Expression (P*eft-3* gene (+NLS) tbb-2UTR) | Cytosolic | Nuclear |
| **Fluorophores** | | | | | | | |
| tagBFP2 | 0.92 | 0 | pCFJ2245 | pCFJ1972 | pMDJ10 | pCFJ2451 | pCFJ1985 |
| ce-gfp | 0.9 | 3 | pCFJ2249 | pCFJ2306 | pMDJ11 | pCFJ2440 | pCFJ2334 |
| mNeonGreen | 0.92 | 0 | pCFJ2262 | pCFJ2308 | pMDJ12 | pCFJ2092 | pCFJ2452 |
| tagRFP-T | 0.95 | 0 | pCFJ2240 | pCFJ2307 | pMDJ13 | pCFJ2437 | pCFJ2442 |
| mCardinal | 0.96 | 0 | pCFJ2241 | pCFJ2309 | pMDJ15 | pCFJ2438 | pCFJ2453 |
| **Photoswitchable** | | | | | | | |
| mMaple3 | 0.97 | 1.5 | pCFJ2210 | pCFJ2301 | pMDJ16 | pCFJ2211 | pCFJ2443 |
| dendra2 | 1 | 1.5 | pCFJ2268 | pCFJ2300 | pMDJ17 | pCFJ2441 | pCFJ2181 |
| **Protein tags** | | | | | | | |
| Halo | 0.97 | 0 | pCFJ2059 | | | | |
| Snap | 0.98 | 0 | pCFJ2060 | | | pCFJ2089 | |
| Clip | 0.98 | 0 | pCFJ2061 | | | pCFJ2088 | |

The genes were codon-adapted for high expression (CAI score)[44], depleted of piRNA homology (piRNA score)[40,45,72], and PATC-rich introns were inserted by Golden-Gate assembly.

(Table 2) for use with arrays or single-copy insertions (e.g., MosSCI or CRISPR tagging) and have deposited the collection with Addgene.

**Efficient genetic engineering with optimized transgenes.** More efficient genetic engineering can accelerate a diverse range of research in laboratories using *C. elegans*. Many genetic engineering techniques (e.g., MosSCI, miniMos, CRISPR/Cas9, and in vivo recombination with FRT or LoxP sites[23–25,64–66]) rely on transient germline expression of injected DNA. Therefore, we reasoned that a set of gene-editing enzymes optimized for consistent and sustained germline expression was likely to improve gene-editing efficiency.

First, we generated optimized, PATC-rich transgenes encoding the *Mos1* transposase (*ce-Mos1*$_{PATC}$) and tested the efficiency of generating MosSCI insertions at one safe harbor insertion site on Chr. V (*oxTi365*)[24]. Injection of P*smu-1::ce-Mos1*$_{PATC}$ generated MosSCI insertions at significantly higher frequency (Fig. 7a) with the highest insertion frequency achieved when using the transgene at 10 ng/ul (Fig. S12).

Transgenic animals carrying arrays are relatively easy to generate compared with single-copy insertions (e.g., MosSCI or CRISPR-tagged alleles). Therefore, it would be advantageous if single-copy insertions could be reliably generated from the continued propagation of a few "founding" array animals. However, single-copy integrations occur almost exclusively in the first few generations[24], presumably owing to progressive transgene silencing. To test if a silencing-resistant *ce-Mos1*$_{PATC}$ could extend this editing window, we co-injected P*smu-1::Mos1*$_{PATC}$ with a miniMos transposon carrying a 6.0 kb P*eft-3::ce-gfp* transgene. We purposely picked array animals with no single-copy insertions segregating in the first three generations to identify insertions generated in later generations insertions. We observed a continuous increase in the number of independent insertions in these transgenic lines until we stopped the experiment after eight generations (Fig. 7b). The optimized P*smu-1::Mos1*$_{PATC}$ also reduced the previously observed strong temperature-dependence of insertion frequency[24]. Generating insertions by simply propagating strains may be an appealing protocol for researchers with limited injection experience and could potentially also be used to generate many independent insertions for large-scale transposon collections (e.g., enhancer or gene traps).

We also generated an optimized *Cas9* transgene (*Cas9*$_{PATC}$) with piRNA homology removed and tested the efficiency for CRISPR-based *gfp* tagging at the endogenous *his-72* locus[25]. A comparison between a commonly used P*eft-3::Cas9* plasmid (pDD133) and P*smu-1::Cas9*$_{PATC}$ showed modest but significantly higher insertion frequency after optimization (Fig. 7c).

Finally, we tested the efficiency of removing a single-copy integrated rescue marker (cbr-*unc-119*(+)) by recombination with optimized *Cre* (P*smu-2::Cre*$_{PATC}$) or enhanced *flp* (P*smu-2::eFlp*$_{PATC}$) recombinases. Both recombinases excised the cassette at high efficiency (~70–80%), quantified by Unc animals on plates two generations after injection (Fig. 7d).

In our laboratory, optimized enzymes with PATCs have consistently improved genetic engineering efficiency and enabled gene-editing for more generations from array animals, presumably by increasing enzyme levels and the duration of germline expression. We have deposited a small collection of optimized gene-editing enzymes at Addgene (Table 3). We propose that including PATCs in enzymes will be a generally useful way to improve the efficiency of current and future gene-editing technologies in *C. elegans*.

**Discussion**
Here, we have generated reagents and determined a set of rules that allow persistent expression of most transgenes in the germline from simple, extrachromosomal arrays. Rule 1: codon-adaption[44], piRNA removal[67], and the addition of introns can improve expression but is rarely sufficient in itself. Rule 2: PATC-rich introns in the coding region improve expression, whereas PATCs in the promoter or 3′-UTR appear to be dispensable. Several PATC-rich introns can be inserted in a single reaction by Golden-gate based cloning or, alternatively, a single intron (intron three from *smu-2*) can be inserted by standard cloning. The shorter 250 bp introns are in most cases preferable although the longer 900 bp introns were better at preventing transgene silencing in repressive chromatin[29]. Rule 3: fluorescent proteins inserted at the C-terminus are less prone to silencing. Rule 4: high transgene concentration (25 ng/ul) enhances expression from simple arrays. Rule 5: two introns, with one intron placed in the first 150 base pairs, stimulate expression. Rule 6: removal of the vector backbone by PCR or restriction digest can prevent silencing of non-optimal transgenes. Rule 7: viral 2A peptides and operons can be used to co-express endogenous genes and

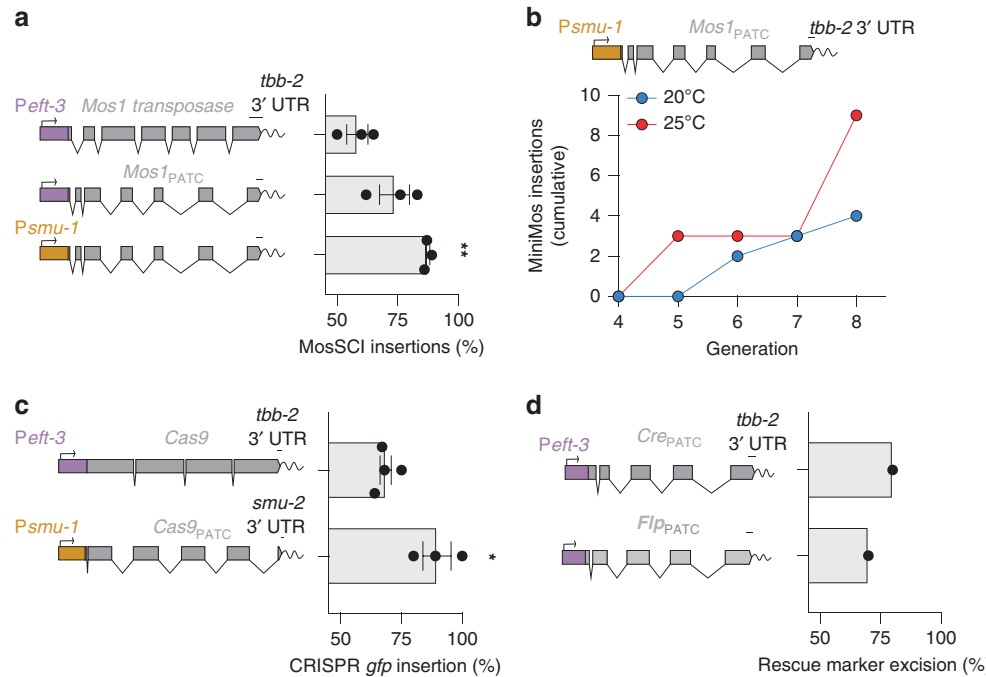

**Fig. 7 PATC-rich transgenes mediate efficient gene-editing. a** MosSCI insertion frequency of a 4.7 kb *gfp* transgene at *oxTi365* (Chr. V) with a codon-optimized Mos1 transposase containing PATCs and using P*eft-3* or P*smu-1*. *n* = 3 biologically independent injections. In all, 20–24 animals were used for each set of injections. Kruskal–Wallis one-way ANOVA. Multiple comparisons: Dunnett's test. **\*\****p* = 0.0063. **b** MiniMos insertion frequency over time. Independent transgenic lines carrying arrays with a P*smu-1*:*mos1*<sub>PATC</sub> transposase and a miniMos transposon with a 6.0 kb transgene were propagated on ten plates each and screened for insertions in every generation. The lines were propagated in parallel at 25 °C (red line) and 20 °C (blue line). *n* = 3 biologically independent transgenic lines. **c** CRISPR/Cas9-mediated GFP tagging of the endogenous *his-72* locus[25] using a Cas9 transgene with no PATCs (P*eft-3*::Cas9) or high PATC content (P*smu-2*::Cas9<sub>PATC</sub>). *n* = 4 and 3 biologically independent injections (top to bottom). For each injection, 9–22 animals were used. Statistics: two-tailed, unpaired *t* test. **\****P* = 0.013. **d** Recombinase excision of rescue markers. Strain with single-copy insertions containing a *cbr-unc-119* rescue cassette flanked by either LoxP or FRT sites were injected with a PATC-rich Cre<sub>PATC</sub> or Flp<sub>PATC</sub> recombinases, respectively. *n* = 1 biologically independent experiments were done for each condition. In all, 10–11 independently injected animals were scored for each experiment. Five lines were propagated for several generations, and the plates were scored for Unc animals with both copies of *cbr-unc-119* removed by recombination. Transgene scale bars = 100 nucleotides. Each datapoint indicates one independent measurement. Bars indicate the mean, and error bars indicate the SEM. Source data are available in the Source Data file.

transgenes. Rule 8: propagating transgenic strains at high temperature (25 °C) allows persistent transgene expression and enables gene-editing for additional generations.

We hope that a description of general transgene engineering rules together with a set of standardized reagents to generate transgenes will facilitate experiments for researchers working on the *C. elegans* germline or those that wish to engineer the genome. In addition, we have aimed to enable research on an enigmatic class of non-coding DNA that has a striking effect on preventing gene silencing in *C. elegans*. Further investigation by us and others will expand upon and reveal mechanisms underlying the resources developed here.

## Methods

**Strains**. *C. elegans* strains were cultured on nematode growth media (NGM) feeding on OP50 or HB101 bacteria and maintained at 15 °C, 20 °C, or 25 °C. *unc-119(ed3)* animals were cultured at 15 to 20 °C on HB101 bacteria, whereas rescued, transgenic array animals were cultured on OP50 bacteria.

**Transgenic animals**. We followed standard protocols for generating strains with extrachromosomal arrays[10] or single-copy insertions by MosSCI, MiniMos, or CRISPR/Cas9[23–25].

Extrachromosomal arrays: we injected into *unc-119(ed3)* animals derived from a 10× outcrossed mutant strain (PS6038) or into the N2 wildtype strain. Selection for arrays was provided by Unc-119 rescue (for plasmids with *cbr-unc-119* in backbone) or antibiotic resistance to hygromycin B[68] using the HygroR plasmid pCFJ782 by adding 500 µl of a 4 mg/ml stock solution (Gold Biotechnology, cat. no. H-270-10) to seeded NGM plates. Every transgenic array line was derived from an independently injected animal.

MosSCI insertions: we inserted single-copy transgenes cloned into pCFJ150 into the universal MosSCI insertion site *oxTi365* on Chr. V by injection into the strain EG8082. The injection mix consisted of 10 ng/ul of the targeting vector pCFJ113 (P*eft-3*::*ce-gfp*::*tbb-2* 3′-UTR) in a pCFJ150 backbone, 10 ng/ul of pCFJ1532 (P*smu-1*::*mosase*<sub>PATC</sub>), 10 ng/ul pCFJ104 (P*myo-3*::*mCherry*), 2.5 ng/ul pCFJ90 (P*myo-2*::*mCherry*), 10 ng/ul pGH8 (P*rab-3*::*mCherry*), 10 ng/ul pMA122 (hs::*peel-1*) and 47.5 ng/ul 1 kb DNA ladder SM1331 (ThermoFisher). We identified insertions by selecting for Unc-119 rescued animals with no fluorescent co-injection markers and no lethality in response to heat-shock expression of the *peel-1* toxin.

MiniMos insertions: we injected 10 ng/ul of a miniMos element (pCFJ1402 - P*eft-3*::*ce-gfp*<sub>PATC</sub>::*tbb-2* 3′-UTR *cbr-unc-119*(+)) into *unc-119(ed3)* animals. The injection mix consisted of 10 ng/ul pCFJ1532 (P*smu-1*::*mosase*<sub>PATC</sub>), 10 ng/ul pCFJ104 (P*myo-3*::*mCherry*), 10 ng/ul pGH8 (P*rab-3*::*mCherry*), 2.5 ng/ul pCFJ90 (P*myo-2*::*mCherry*) and 57.5 ng/ul 1 kb DNA ladder SM1331 (ThermoFisher). The backbone of pCFJ1402 contains the negative selection marker hs::*peel-1* to kill animals with extrachromosomal arrays in response to heat-shock. To test for insertions across generations, we picked three independent transgenic array lines that did not segregate miniMos insertions in the first three generation and propagated the lines at 20 °C and 25 °C on 10 plates each. For every generation, we transferred approx. ten animals to new plates before heat-shocking starved plates to identify miniMos insertions based on the lack of fluorescent co-injection markers. Any plate that gave an insertion was not propagated for more generations to avoid counting any transposon insertion twice.

CRISPR/Cas9 insertions: we tagged the *his-72* locus with *gfp*. We generated a 4.5 kb repair template (pMNK17) derived from pDD129[25] with 400 bp homology regions and *cbr-unc-119*(+) rescue. Importantly, the repair template does not contain the *his-72* promoter and no GFP fluorescence is observed prior to successfully tagging the endogenous *his-72* locus. A single guide RNA with the spacer 5′-AGCTTAAGCACGTTCTCCG-3′ was expressed from a plasmid (pMNK18) using a U6 promoter. We expressed Cas9 from plasmids pCFJ1646 (P*eft-3*::Cas9) or pCFJ2474 (P*smu-2*::Cas9<sub>PATC</sub>::*sl2*::*tagRFP*). The injection mix consisted of 25 ng/ul of the Cas9 plasmid (pCFJ1646 or pCFJ2474), 10 ng/ul of the repair template (pMNK17), 25 ng/ul of the sgRNA (pMNK18), and 40 ng/ul 1 kb

**Table 3 Optimized gene-editing enzymes.**

| Characteristics | | | Optimized enzyme | | Germline optimized (with PATCs) | | | |
|---|---|---|---|---|---|---|---|---|
| Enzyme | CAI score | piRNA score | Gateway entry vector | Expression (P*eft-3*) | Gateway entry vector | Expression (P*smu-1*) | RFP co-expression (P*smu-2* tagRFP) | Inducible expression (P*hsp*-16.41) |
| Mos1 transposase | 0.92 | 0 | pCFJ1470 | pCFJ1477 | pCFJ2471 | pCFJ1532 | pCFJ2475 | |
| Cre recombinase | 0.99 | 0 | pCFJ2118 | pCFJ2139 | pCFJ2126 | pMDJ20 | pMDJ22 | pMDJ113 |
| Enhanced Flp recombinase | 0.7 | 1.5 | pCFJ2119 | pCFJ2130 | pCFJ2124 | pMDJ21 | pMDJ23 | pMDJ112 |
| Cas9 | 0.98 | 9 | pCFJ2469 | | pCFJ2470 | pCFJ2477 | pCFJ2474 | |

The genes were codon-adapted for high expression (CAI score)[44], depleted of piRNA homology (piRNA score)[40,45,72], and PATC-rich introns were inserted by Golden-Gate assembly.

DNA ladder SM1331 (ThermoFisher). We injected this mix into *unc-119*(*ed3*) animals and identified insertions based on Unc-119 rescue and ubiquitous GFP expression, including the germline.

All injection strains are available from the Caenorhabditis elegans Genetics Center (CGC).

**Imaging**. All measurements were taken from distinct samples (defined as an independently generated transgenic animal), except for time-course measurements where the same sample was measured repeatedly every two generations. The sample sizes and all primary data (percentage of animals with fluorescent germline) are included in Source Data. Transgenic animals were generated and imaged in a stereotyped way, as described below, to ensure consistency. In all, 1–2 injected animals were placed on individual NGM plates seeded with HB101 at 25 °C in a temperature-controlled incubator. Plates were allowed to starve out and inspected for rescued F2 progeny, indicating that a plate contained stable transgenic lines. Such plates were "chunked" to a new plate, and a single young F2 adult animal with eggs was picked two days later, ensuring that only a single independent line was picked from any injected animal. Three days later, the F3 progeny of this clonal animal was scored for germline fluorescence by mounting animals on agarose pads (2%) and anesthetizing the animals with 50 mM sodium azide. We imaged animals on upright, non-motorized compound microscopes (Leica DM2500 and Zeiss Axioimager Z.2) with ×42 or ×60 oil immersion objectives and scored germline fluorescence in 11 animals from each independent strain. Both gonad arms were scored for GFP fluorescence and every animal was quantified in a binary way ("on" or "off"). The experimenter was not blinded to the genotype of the transgenic animals.

**Molecular biology**. Molecular biology was performed using standard protocols and commercial available reagents. A step-by-step protocol describing the Golden-Gate-based method for inserting PATC-rich introns into a synthetic transgene can be found at Protocol Exchange[69]. All reactions were designed using the free molecular biology editor "A plasmid Editor" (ApE) developed and maintained by M Wayne Davis. Annotated DNA sequences for all plasmids are included in the Source Data file. All plasmids are available upon request from Addgene or from the corresponding author.

**Statistical analysis**. The statistical analysis was performed using GraphPad Prism v8 for macOS. The specific tests performed are listed in the legends of individual figures, and the primary data for every figure is included in the Source Data file. In general, fluorescence expression is stochastically silenced with frequent "all or none" observations (i.e., complete silencing or full expression), and the data do not follow a Gaussian distribution. Therefore, most of the utilized statistical tests are non-parametric tests.

**Online analysis**. The website www.wormbuilder.org/PATC/ was written in R programming language. Its online execution occurs through an Amazon Web Services (AWS) Elastic Computing Cloud (EC2) instance. The source code can be obtained at: https://github.com/AmhedVargas/PATC_2_0. Please see "Software and code" in the accompanying Reporting Summary for detailed information on all software used, including version numbers.

**Reporting summary**. Further information on research design is available in the Nature Research Reporting Summary linked to this article.

## Data availability

All data generated or analyzed during this study are included in this published article (and its supplementary information files). Any other relevant data are available from the authors upon reasonable request. Source data are provided with this paper.

## Code availability

The source code[70] for www.wormbuilder.org/PATC/ is available at: https://github.com/AmhedVargas/PATC_2_0. Please see Supplementary Methods and the Reporting Summary for detailed information on all software used, including version numbers.

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

## Acknowledgements

We thank Andrew Z. Fire and Erik M. Jorgensen for experimental support, M. Wayne Davis for bioinformatic assistance, and Kam Hoe for technical assistance. Some strains were provided by the CGC, which is funded by NIH Office of Research Infrastructure Programs (P40 OD010440). We thank the KAUST Bioscience Core Labs and the Linux and Advanced Platforms team for expert assistance and Faisal Alkhaldi for assistance developing www.wormbuilder.org. This work was funded by a KAUST intramural grant and the funders had no role in study design, data collection and analysis, decision to publish, or preparation of the manuscript.

## Author contributions

Christian Frøkjær-Jensen: conceptualization, methodology, formal analysis, investigation, resources, writing—original draft, writing—review & editing, visualization, funding acquisition. Mohammed D. Aljohani: investigation, writing—review & editing. Sonia El Mouridi: methodology, investigation, writing—review & editing, visualization. Monika Priyadarshini: investigation, writing—review & editing. Amhed M. Vargas-Velazquez: software, data curation, writing—review & editing, visualization.

## Competing interests

The authors declare no competing interests.
