## [Peer Review File · Nature Communications]

Reviewers' Comments:

Reviewer #1:

Remarks to the Author:

Over a decade ago, Andy Fire observed that genes normally expressed in the germline of *C. elegans* contained regulatory sequences and introns with abundant "PATCs," (periodic An/Tn clusters), stretches of DNA sequence containing runs of As and Ts at 10-bp intervals. Such periodic clusters were also found in related nematodes. Subsequent work by Frokjaer-Jensen et al. revealed that such DNA sequences have an "anti-silencing" effect on transgene expression in the germline, and work from other authors has apparently shown they may be effective at preventing transgene silencing in human liver cells as well. Here, the authors have built on that work by developing a new web-based computational tool to identify PATCs, investigating where and how to use PATCs in transgene constructs to achieve reliable germline expression in *C. elegans*, and developing a cloning strategy to insert PATC-rich introns into transgenes of interest. In principle this is an important set of accomplishments that may enhance research from many groups by making this anti-silencing approach more accessible and robust. It does not move the field fundamentally beyond the prior work of the senior author, but it useful tools for and some new insights into the use of PATCs for germline expression.

Unfortunately, I frequently found the presentation of the work in this manuscript to be frustrating, and the logic of the experiments to be unnecessarily opaque. For this work to be of maximal value, it's critical that the conclusions and recommendations are clear to readers who are unfamiliar with the lengthy history of work on germline expression and silencing in *C. elegans*. The manuscript requires extensive editing for accuracy and clarity, ideally with a relatively naïve reader (e.g., a 2nd or 3rd year graduate student) in mind. I think that if the results are presented in a more orderly and precise way, they will be of significant use to the community – indeed, they are already in use by the community, thanks to the generosity of the senior author in sharing unpublished reagents. Many of the problems can probably be addressed through careful editing. I leave it to the editors to determine whether they are severe enough to warrant re-review.

The introduction should better summarize prior knowledge about the key phenomena relevant to this work: First, that transgene silencing remains a somewhat mysterious phenomenon and a major experimental obstacle in the *C. elegans* germline (it is not clear as written that this phenomenon is largely restricted to the germline, or why this might be). It should succinctly summarize what is known about the silencing mechanisms – e.g., it shows copy-number and other context-specific effects and involves dsRNA intermediates. Are the effects thought to be primarily transcriptional, posttranscriptional, or some combination? How might these effects be ameliorated by PATCs and/or by elevated temperature? To my knowledge there is little evidence that PATCs are prevalent or protect genes from silencing in organisms other than *C. elegans* and perhaps some closely related species. The manuscript should also be clearer in its statements about the potential conservation of the silencing phenomena and the role of PATCs in other organisms, including close relatives of *C. elegans*, as well as other experimental organisms.

One minor concern about terminology that I think is confusing but could easily be addressed: prior studies, as well as the current paper, have shown that the expression of transgenes in the *C. elegans* germline is often silenced under standard culture conditions (20°C) but can be desilenced at elevated temperature (25°C). In my view, this should not be described (as it is here) as "temperature-sensitive" expression (p.8) or "temperature-dependent silencing," since expression is stronger at higher temperature – it could be described as temperature-dependent expression or cold-sensitive silencing.

pp. 5-6 – I find the conclusions from these *smu-2* expression experiments to be confusing. Is it fair to conclude that the introns in *smu-2*, together with the PATC-rich promoter, enable expression from arrays? This should be stated more clearly, since it is one of the key messages of the work. Later (p. 7) it is stated that "PATCs in the promoter or introns stimulate expression," but

this does not seem to accurately summarize the findings, since PATCs in the promoter are apparently insufficient to ensure expression (Figure 3C), so I think it would be more precise to say that "PATCs in the promoter and introns contribute additively to protecting transgenes from silencing, while those in the 3' UTR seem to have little effect."

How do the authors conclude that "Codon-optimization of transgenes is not necessary for germline expression but shows synergy with incorporation of PATCs"? First, the term "synergy" indicates that the effects are more than additive, which I don't think is the case here (or at least has not been shown to be true). Second, it does not seem that codon-optimization was explicitly tested here, since the two GFP genes compared in this work differ beyond their codon optimization, so I think this statement should be revised. In my view, the evidence that codon optimization promotes protein expression in *C. elegans* is very limited, and seems to be based largely on a single study (Redemann et al. 2011) of a single gene. The widely used codon optimization algorithm used in that paper takes the simple approach of using the most abundant codon for each amino acid as a default, and does not generate sequences that resemble the codon usage of naturally occurring *C. elegans* genes. The effects of different methods of codon optimization have not been tested systematically either here or elsewhere.

The important paragraph on p. 6 beginning with "We turned to fluorophores" is particularly problematic. I tried to rewrite it but in doing so I realized that I did not understand several statements within this paragraph. Some of the conclusions in this paragraph seem ill-founded – in particular, the last part of the paragraph states that when PATC-rich promoters and 3' UTRs from *C. briggsae* were used, almost no germline expression was seen, but that "This lack of expression was not due to an inactive promoter as PATC-rich introns from *C. elegans* boosted expression from the *C. briggsae* promoters." I don't see how these observations are internally consistent. The conclusion: "inclusion of non-coding endogenous DNA sequences significantly increases the frequency and reliability of expression" is neither very clear nor very helpful – which regulatory sequences (promoter, introns, 3'UTR) actually matter?

For the first part of this paragraph I suggest rewriting as follows:

"To define general rules for anti-silencing by PATCs, we designed a coding sequence for green fluorescent protein (GFP) that does not contain any homology to endogenous coding sequences or known piRNAs, which can silence germline genes. This synthetic gene also lacks homology to 22G RNAs, which are thought to protect genes from silencing through a pathway dependent on the Argonaute protein CSR-1. This coding sequence was designed using a popular web-based platform for *C. elegans* codon adaptation (Redemann et al., 2011), and a custom algorithm to eliminate sequences homologous to known piRNA sequences (Batista et al., 2008; Bagijn et al., 2012; Lee et al., 2012). The construct also contained several [three?] small, synthetic introns that were designed to enable subsequent insertion of large stretches of PATC-rich sequence. When this "ce-GFP" gene was inserted between the promoter and 3' UTR from the *smu-1* gene, GFP was robustly expressed. In contrast, a more conventional GFP gene derived from the natural *A. victoria* sequence was frequently silenced (Figure 3C). Insertion of PATC-rich introns further improved the expression of the ce-GFP gene; this enhancement was more obvious when the more silencing-prone promoter from *pie-1* was used (Figure 3D).

[I do not understand the rest of the paragraph well enough to rephrase it clearly]

I also find it very confusing that the work first describes expression from simple arrays and then abruptly shifts to experiments using complex arrays to assay expression (bottom of p. 6). The rationale for using complex arrays is unclear to me, since PATCs appear to allow expression from simple arrays, which are easier to generate. Additionally, the authors found that expression shows a positive correlation with the concentration of the transgene used in creating complex arrays, so is it actually helpful to make complex arrays? Ideally, the authors should directly compare complex to simple arrays, since the latter require less effort to construct, and are probably more reproducible, since different labs use different sources of genomic "carrier" DNA. If it is possible to get robust germline expression at 20°C from simple arrays using a strong, PATC-rich germline

promoter and introns, is there any reason to use complex arrays?

Also, why would investigators use the *pie-1* or *mex-1* promoter if the *smu-1* and *smu-2* promoters are more reliable? Can a transgene using the [*smu-2* promoter + PATC-rich introns + a germline 3' UTR] recapitulate the expression pattern of the gene from which the 3' UTR is obtained? If so, this seems like it would be a simplest, robust way to deploy PATCs for germline expression.

I wish the authors had compared other parameters, such as the effects of different numbers of introns and their position within the gene of interest. Does it matter if the introns are towards the 5' or 3' end of the transgene? If a coding sequence of interest is fused to a fluorescent protein with PATC-rich introns, is this sufficient for expression? Does this work if the fluorescent protein coding sequence is fused to either end of the gene of interest? While I don't think they are essential for publication at this time, I do feel that addressing these issues would elevate the impact of the work. For example, if a gene can be expressed when inserted into a vector containing the *smu-1* or *smu-2* promoter upstream of a fluorescent protein with PATC-rich introns, and an appropriate germline 3' UTR, this would greatly simplify researcher's decisions about how to apply these findings.

In quite a few cases throughout the work, the authors' experiments do not appear to have been designed in a way that enables them to directly test the importance of specific gene features for germline expression – specifically, they compare transgenes that have several major differences and conclude that one of them is important. They also frequently overinterpret their findings. For example, in Figure 6A the authors compare 3 different constructs expressing the *Mos1* transposase (a.k.a. "mosase") and conclude that more PATC-rich introns confer more efficient insertion of mini-Mos. However, the most efficient construct also used a different promoter and 3' UTR than the other two they assayed, making it impossible to attribute elevated or more consistent expression specifically to the introns. Similarly, they conclude that PATCs improve the expression/activity of Cas9 for tagging endogenous loci, but their constructs have different promoters (the 3' UTRs are not shown, and may also be different). They also provide numerous potential introns, which can be ordered through Addgene, but they don't systematically compare their performance, so how should the reader decide what to use? I do not think it is necessary that the authors fill in all the gaps in their analysis, which would have only limited value and would unnecessarily delay publication, but it is important that they edit the manuscript carefully to avoid making any claims that are not supported by their data. I have highlighted some specific cases of overinterpretation. but they need to do this systematically throughout the manuscript.

All reported experimental results should indicate (in the figures and/or legends) whether they were obtained using simple or complex arrays, and at what temperature expression was assayed.

It would be helpful in some cases for the authors to measure expression using a more quantitative approach than a simple binary on/off assay. Investigators who want to use these tools would benefit from more information about how the context of the transgene (single-copy, simple or complex array, etc.) influences expression levels.

The creation of lines expressing genome editing enzymes (Cas9, *Mos* transposase, and Cre and Flp recombinases) may be quite valuable to the community – the latter 2 may facilitate the RMCE method recently published by Michael Nonet (which the authors may wish to cite).

Minor corrections/suggestions (not comprehensive, just things I noted):

Remove the extra comma after "simple" in p.3 line 10

p. 4" "Our first aim was to enable broadly deployable identification of PATCs." I suggest: "We developed a user-friendly, versatile web server to identify and quantify PATCs."

p.7, first full paragraph "refractory" is a better term in this context than "recalcitrant"

Lines 25-26: "...as well as a fundamental lack of rules for how to best generate silencing-resistant transgenes." Suggest: "...lack of clarity about best practices for creating silencing-resistant transgenes"

I'm not sure whether Figure 1 is useful – I'm not sure whether it would be printed/displayed in a way that the text would be readable, so a callout to the actual web interface may be more useful.

"A comparison between a commonly used Peft-3:Cas9 plasmid (pDD133, (Dickinson et al., 2013)) and a similar PATC-rich Cas9 construct (Psmu-1:Cas9(PATC+)) showed higher insertion frequencies, in some cases more than double, for three researchers with different levels of experience (Figure 6C)." What is the conclusion of this analysis? It's not clear how these plasmids were compared or which construct actually worked better.

I don't understand the following statements:

"We note that the requirement for PATCs internal to the transgenes itself is in line with the proposed biological role for PATCs; if insertion of foreign DNA into a PATC-rich genomic environment alone conferred resistance to silencing then it would be difficult to imagine how An/Tn sequences could participate in a genome defense."

What are "PATCs internal to the transgene itself..."? and what are the implications for the mechanism of PATC-mediated anti-silencing?

Reviewer #2:

Remarks to the Author:

In this manuscript by Aljohani et al, the authors set out to develop a method for transgene expression in *C. elegans* from extrachromosomal arrays that overcomes transcriptional silencing. Extrachromosomal arrays are a common method for expression of transgenes in somatic *C. elegans* tissues. However, arrays are silenced in the germline, which has limited their usefulness in this important tissue. Germline silencing is not unique to *C. elegans* and likely reflects mechanisms designed to protect the germline from invaders, such as transposons.

The authors show that the addition of Periodic An/Tn Clusters (PACTs) and removal of piRNA recognition sequences results in robust and stable transgene expression in the germline from extrachromosomal arrays. Moreover, addition of PACTs improves the efficiency of CRISPR-Cas9 edits and transposon-mediated single-copy insertions by improving genomic editing in the germline.

Overall the data is of high quality and the methods and tools developed here will be broadly useful to the *C. elegans* community. There is also potential for the information provided here to influence analysis of germline silencing in other species. I strongly support publication of this paper and list below a few minor issues that the authors should address in their revision.

Comments:

- Throughout the manuscript the authors use a specific nomenclature to refer to the different GFP versions that they used (standard GFP, ce-GFP and ce-GFP(PACT)). However, I was confused by some of the versions that they show in the supplement. For instance, what is wGFP? (Fig. S1C). In Figure S2B, was ce-GFP used? (it contains two syntons, whereas standard GFP has three). In Figure S4A, why does ce-GFP contain four syntons, while in the rest of the paper it only contains two?

- Another source of confusion was the data in figure S2B. The authors show robust expression of

GFP from the *smu-1* promoter using a variety of 3'UTRs, which contradicts the results from Fig. 3C, which shows no expression of standard GFP under similar conditions (unless ce-GFP was used in fig. S2B, in which case they need to specify this).

- In Figure 3B, why does GFP localize to the nuclei in the germline?

- In Figures 5A and 5B, it is very hard to see the signals that the arrows are pointing to. It might be better to show these in grayscale instead of green. It might also help to zoom in to the germline region in Fig. 5A.

- In page 5, the authors say that their method for Golden Gate Cloning will be detailed in the Supplementary Protocol, but this is missing from the files provided.

- The legend for Fig. S2 that is immediately below the figure is incorrect.

- Figure S2A: Can the authors comment on why inserting GFP at the N-terminus results in silencing?

Response to referees.

Author replies are highlighted in blue.

We would like to thank the reviewers for their careful reading of our manuscript and their constructive comments to improve the manuscript. We have implemented most of the suggested changes and have also included a substantial amount of new data to strengthen and expand the conclusion of the paper. We believe the manuscript has substantially improved as a result of the review process.

Reviewer #1 (Remarks to the Author):

Over a decade ago, Andy Fire observed that genes normally expressed in the germline of *C. elegans* contained regulatory sequences and introns with abundant “PATCs,” (periodic An/Tn clusters), stretches of DNA sequence containing runs of As and Ts at 10-bp intervals. Such periodic clusters were also found in related nematodes. Subsequent work by Frokjaer-Jensen et al. revealed that such DNA sequences have an “anti-silencing” effect on transgene expression in the germline, and work from other authors has apparently shown they may be effective at preventing transgene silencing in human liver cells as well. Here, the authors have built on that work by developing a new web-based computational tool to identify PATCs, investigating where and how to use PATCs in transgene constructs to achieve reliable germline expression in *C. elegans*, and developing a cloning strategy to insert PATC-rich introns into transgenes of interest. In principle this is an important set of accomplishments that may enhance research from many groups by making this anti-silencing approach more accessible and robust. It does not move the field fundamentally beyond the prior work of the senior author, but it useful tools for and some new insights into the use of PATCs for germline expression.

We thank the reviewer for the perceived usefulness of the manuscript.

Unfortunately, I frequently found the presentation of the work in this manuscript to be frustrating, and the logic of the experiments to be unnecessarily opaque. For this work to be of maximal value, it’s critical that the conclusions and recommendations are clear to readers who are unfamiliar with the lengthy history of work on germline expression and silencing in *C. elegans*. The manuscript requires extensive editing for accuracy and clarity, ideally with a relatively naïve reader (e.g., a 2nd or 3rd year graduate student) in mind. I think that if the results are presented in a more orderly and precise way, they will be of significant use to the community – indeed, they are already in use by the community, thanks to the generosity of the senior author in sharing unpublished reagents. Many of the problems can probably be addressed through careful editing. I leave it to the editors to determine whether they are severe enough to warrant re-review.

We thank the reviewer for the constructive suggestions on how to improve the clarity of the manuscript. Much thought and effort obviously went into improving the manuscript. We have revised the manuscript substantially to make it as accessible as possible for

researchers at all levels. In several cases, the paragraphs are changed so specific recommendations may no longer be relevant. In most other cases, we have incorporated the suggested changes more or less verbatim.

The introduction should better summarize prior knowledge about the key phenomena relevant to this work: First, that transgene silencing remains a somewhat mysterious phenomenon and a major experimental obstacle in the *C. elegans* germline (it is not clear as written that this phenomenon is largely restricted to the germline, or why this might be). It should succinctly summarize what is known about the silencing mechanisms – e.g., it shows copy-number and other context-specific effects and involves dsRNA intermediates. Are the effects thought to be primarily transcriptional, posttranscriptional, or some combination? How might these effects be ameliorated by PATCs and/or by elevated temperature? To my knowledge there is little evidence that PATCs are prevalent or protect genes from silencing in organisms other than *C. elegans* and perhaps some closely related species. The manuscript should also be clearer in its statements about the potential conservation of the silencing phenomena and the role of PATCs in other organisms, including close relatives of *C. elegans*, as well as other experimental organisms.

We agree that the manuscript could be structured as suggested here. However, we have opted for a shorter and more concise introduction that highlights silencing mechanisms in mammals and plants before describing the key observations in *C. elegans*. We are already at the limit of references allowed by *Nature Communications* (70 references) and we therefore do not think an expanded introduction is feasible. From a practical point of view, we believe we supply enough information for a broad audience to utilize the tools and reagents that form the core of the manuscript.

One minor concern about terminology that I think is confusing but could easily be addressed: prior studies, as well as the current paper, have shown that the expression of transgenes in the *C. elegans* germline is often silenced under standard culture conditions (20°C) but can be desilenced at elevated temperature (25°C). In my view, this should not be described (as it is here) as “temperature-sensitive” expression (p.8) or “temperature-dependent silencing,” since expression is stronger at higher temperature – it could be described as temperature-dependent expression or cold-sensitive silencing.

We have changed the terminology to avoid confusion and now use the term “temperature-dependent” expression. We are aware of the description by Strome et al. (2001) that describes how a shift from 16°C to 25°C resulted in partial re-activation of germline expression (in 7 of 22 animals). Their experiments differ from our experimental conditions in that persistent germline fluorescence required continuous, visual selection for “expressor” lines. In contrast, we purposefully did not apply any selective pressure therefore think the data is important to include. We have expanded the data with several more transgenic lines showing the same reversible fluorescence silencing.

pp. 5-6 – I find the conclusions from these smu-2 expression experiments to be

confusing. Is it fair to conclude that the introns in *smu-2*, together with the PATC-rich promoter, enable expression from arrays? This should be stated more clearly, since it is one of the key messages of the work. Later (p. 7) it is stated that “PATCs in the promoter or introns stimulate expression,” but this does not seem to accurately summarize the findings, since PATCs in the promoter are apparently insufficient to ensure expression (Figure 3C), so I think it would be more precise to say that “PATCs in the promoter and introns contribute additively to protecting transgenes from silencing, while those in the 3' UTR seem to have little effect.”

Agreed. We have modified this section of the manuscript substantially. Our new data show that PATC-rich introns stimulate germline expression whereas PATCs in the promoter and 3' UTR have no consistent effect. We have clarified this in the interim conclusion on page 7:

"We have found no evidence that PATC-rich promoters or 3' UTRs improve germline expression."

How do the authors conclude that “Codon-optimization of transgenes is not necessary for germline expression but shows synergy with incorporation of PATCs”? First, the term “synergy” indicates that the effects are more than additive, which I don’t think is the case here (or at least has not been shown to be true). Second, it does not seem that codon-optimization was explicitly tested here, since the two GFP genes compared in this work differ beyond their codon optimization, so I think this statement should be revised. In my view, the evidence that codon optimization promotes protein expression in *C. elegans* is very limited, and seems to be based largely on a single study (Redemann et al. 2011) of a single gene. The widely used codon optimization algorithm used in that paper takes the simple approach of using the most abundant codon for each amino acid as a default, and does not generate sequences that resemble the codon usage of naturally occurring *C. elegans* genes. The effects of different methods of codon optimization have not been tested systematically either here or elsewhere.

We agree and have included the following sentences to clarify:

"Codon-optimization modestly increased the frequency of expression but did not reach statistical significance (Fig. 2a)." (page 6)

*"We tested the modest effect of codon-optimization using the *smu-1* promoter (Fig. 2b). With *Psmu-1*, we observed substantial germline array expression of *ce-gfp* with synthetic introns and a further increase from adding PATC-rich introns (Fig. 2b). A codon-optimized *mCherry* (*ce-mCherry*) with synthetic introns was poorly expressed using *Psmu-1*, but PATC-rich introns significantly increased expression (Fig. 2c). Similarly, a *smu-1* promoter from *C. briggsae* (*Pcbr-smu-1*) also required addition of PATCs for robust germline expression (Fig. S4). These results demonstrate that codon-optimization in itself does not necessarily ensure germline expression." (page 6)*

"Rule 1: codon-adaption⁴⁴, piRNA removal⁶⁸, and the addition of introns can improve expression but is rarely sufficient in itself." (page 13).

The important paragraph on p. 6 beginning with “We turned to fluorophores” is particularly problematic. I tried to rewrite it but in doing so I realized that I did not understand several statements within this paragraph. Some of the conclusions in this paragraph seem ill-founded – in particular, the last part of the paragraph states that when PATC-rich promoters and 3' UTRs from *C. briggsae* were used, almost no germline expression was seen, but that “This lack of expression was not due to an inactive promoter as PATC-rich introns from *C. elegans* boosted expression from the *C. briggsae* promoters.” I don't see how these observations are internally consistent. The conclusion: “inclusion of non-coding endogenous DNA sequences significantly increases the frequency and reliability of expression” is neither very clear nor very helpful – which regulatory sequences (promoter, introns, 3'UTR) actually matter?

Yes, we agree that this paragraph and section was particularly problematic. We have revised that section of the manuscript extensively (including adding a substantial amount of data) to clarify that the main determinant of germline expression is PATC content in introns.

For the first part of this paragraph I suggest rewriting as follows:

“To define general rules for anti-silencing by PATCs, we designed a coding sequence for green fluorescent protein (GFP) that does not contain any homology to endogenous coding sequences or known piRNAs, which can silence germline genes. This synthetic gene also lacks homology to 22G RNAs, which are thought to protect genes from silencing through a pathway dependent on the Argonaute protein CSR-1. This coding sequence was designed using a popular web-based platform for *C. elegans* codon adaptation (Redemann et al., 2011), and a custom algorithm to eliminate sequences homologous to known piRNA sequences (Batista et al., 2008; Bagijn et al., 2012; Lee et al., 2012). The construct also contained several [three?] small, synthetic introns that were designed to enable subsequent insertion of large stretches of PATC-rich sequence. When this “ce-GFP” gene was inserted between the promoter and 3' UTR from the *smu-1* gene, GFP was robustly expressed. In contrast, a more conventional GFP gene derived from the natural *A. victoria* sequence was frequently silenced (Figure 3C). Insertion of PATC-rich introns further improved the expression of the ce-GFP gene; this enhancement was more obvious when the more silencing-prone promoter from *pie-1* was used (Figure 3D).

[I do not understand the rest of the paragraph well enough to rephrase it clearly]

We thank the reviewer for the excellent re-phrasing and have incorporated the suggestion almost verbatim.

I also find it very confusing that the work first describes expression from simple arrays and then abruptly shifts to experiments using complex arrays to assay expression (bottom of p. 6). The rationale for using complex arrays is unclear to me, since PATCs appear to allow expression from simple arrays, which are easier to generate. Additionally, the authors found that expression shows a positive correlation with the concentration of the transgene used in creating complex arrays, so is it actually helpful

to make complex arrays? Ideally, the authors should directly compare complex to simple arrays, since the latter require less effort to construct, and are probably more reproducible, since different labs use different sources of genomic “carrier” DNA. If it is possible to get robust germline expression at 20°C from simple arrays using a strong, PATC-rich germline promoter and introns, is there any reason to use complex arrays?

This concern rests on a misunderstanding. We did not test complex arrays - all of the data shown is from simple arrays containing only plasmid and 1 kb ladder DNA as a stuffer. Expression from simple arrays is stimulated by higher transgene concentrations. To avoid confusion, we have clarified through-out the manuscript that we are only using simple arrays.

Also, why would investigators use the *pie-1* or *mex-1* promoter if the *smu-1* and *smu-2* promoters are more reliable?

The *pie-1* and *mex-5* promoters are useful because they are germ-cell specific. In another case, we have used the sub-optimal *pie-1* promoter in order to be in a "dynamic" range where we could detect improved expression by changing some aspect of the transgene.

Can a transgene using the [*smu-2* promoter + PATC-rich introns + a germline 3' UTR] recapitulate the expression pattern of the gene from which the 3' UTR is obtained? If so, this seems like it would be a simplest, robust way to deploy PATCs for germline expression.

We have not tested this specific experimental condition. The reason for using *Pmex-5* to determine the role of 3' UTRs in regulating germline expression is that the microscopy is easier when there is no expression in somatic tissue (where *Psmu-1* and *Psmu-2* are also expressed). And germline expression from *Pmex-5* using a PATC-rich *gfp* (*ce-gfp_{PATC}*) is quite robust.

I wish the authors had compared other parameters, such as the effects of different numbers of introns and their position within the gene of interest. Does it matter if the introns are towards the 5' or 3' end of the transgene? If a coding sequence of interest is fused to a fluorescent protein with PATC-rich introns, is this sufficient for expression? Does this work if the fluorescent protein coding sequence is fused to either end of the gene of interest? While I don't think they are essential for publication at this time, I do feel that addressing these issues would elevate the impact of the work. For example, if a gene can be expressed when inserted into a vector containing the *smu-1* or *smu-2* promoter upstream of a fluorescent protein with PATC-rich introns, and an appropriate germline 3' UTR, this would greatly simplify researcher's decisions about how to apply these findings.

We agree with the reviewer and have added a substantial amount of new data to explore these parameters:

Figure 2d-e and Figure S5 test how many introns are required to enhance expression (sometimes, only one).

Figure 3 shows that at least two introns are required for germline expression from arrays, with a requirement for one intron within the first 150-250 nucleotides for "full" expression.

Figure 4 demonstrates that a gene with PATC-rich introns can be used to co-express a transgene using 2A viral peptides and operons.

Figure 2f shows the importance of tagging transgenes at the C-terminus, if possible.

We have also structured the first part of the discussion to highlight a set of "rules" that improve germline expression.

These results should, as the reviewer states, facilitate experimental choices for other researchers.

In quite a few cases throughout the work, the authors' experiments do not appear to have been designed in a way that enables them to directly test the importance of specific gene features for germline expression – specifically, they compare transgenes that have several major differences and conclude that one of them is important.

We agree and have generated a new set of data to address this shortcoming. Most importantly, we have added data to figure 2a which specifically tests the role of PATCs in introns without changing the structure of the transgene. Importantly, we see a significant increase in germline expression from inserting 250 bp or 900 bp introns with PATCs. Size-matched endogenous introns with no PATCs do not enhance germline expression.

They also frequently overinterpret their findings. For example, in Figure 6A the authors compare 3 different constructs expressing the Mos1 transposase (a.k.a. "mosase") and conclude that more PATC-rich introns confer more efficient insertion of mini-Mos. However, the most efficient construct also used a different promoter and 3' UTR than the other two they assayed, making it impossible to attribute elevated or more consistent expression specifically to the introns. Similarly, they conclude that PATCs improve the expression/activity of Cas9 for tagging endogenous loci, but their constructs have different promoters (the 3' UTRs are not shown, and may also be different).

We have rephrased our conclusions and are now merely state that, in our hands, "optimized" gene editing enzymes are very efficient. And in some cases better than commonly used reagents. Comparison of efficiencies between laboratories is notoriously difficult. We have deposited the reagents as a single "plasmid kit" at Addgene. It will, therefore, be relatively easy for every laboratory to test the relative efficiency in their hand.

They also provide numerous potential introns, which can be ordered through Addgene, but they don't systematically compare their performance, so how should the reader decide what to use?

We have tested 250 bp and 900 bp introns as well as a single intron from smu-2 (intron #3). All of the introns stimulate robust germline expression from arrays. We previously tested the 250 bp and 900 bp introns as single-copy insertions into repressive chromatin (Froekjaer-Jensen et al. 2016). In that specific condition, the larger introns were slightly more efficient and we have therefore included them in the "kit". However, for practical purposes the shorter 250 bp introns are likely the most useful.

We have added the following sentence to ease the reader's decision:

"The shorter 250 bp introns are in most cases preferable although the longer 900 bp introns were better at preventing transgene silencing in repressive chromatin²⁹."

I do not think it is necessary that the authors fill in all the gaps in their analysis, which would have only limited value and would unnecessarily delay publication, but it is important that they edit the manuscript carefully to avoid making any claims that are not supported by their data. I have highlighted some specific cases of overinterpretation. but they need to do this systematically throughout the manuscript.

We have carefully edited the manuscript to avoid over-interpreting the data.

All reported experimental results should indicate (in the figures and/or legends) whether they were obtained using simple or complex arrays, and at what temperature expression was assayed.

We have added this information to the legends and the methods. Also, Supplementary Table 1 contains all the primary data including a description of temperature and that all arrays are "simple".

It would be helpful in some cases for the authors to measure expression using a more quantitative approach than a simple binary on/off assay. Investigators who want to use these tools would benefit from more information about how the context of the transgene (single-copy, simple or complex array, etc.) influences expression levels.

Quantifying absolute expression levels is time-consuming and expression from arrays is highly variable. Changing the method for quantifying expression would have precluded testing such large numbers of transgenes and independent animals. We believe our study has generated and tested an order of magnitude more transgenes and independent transgenic lines than any other study in the field.

We refer the reviewer to Froekjaer-Jensen et al (2016) for a quantification of absolute somatic expression levels in L1 animals carrying single-copy insertions, where we demonstrated that PATCs do not appear to influence the level of expression.

The creation of lines expressing genome editing enzymes (Cas9, Mos transposase, and Cre and Flp recombinases) may be quite valuable to the community – the latter 2 may facilitate the RMCE method recently published by Michael Nonet (which the authors may wish to cite).

Thank you for the suggestion. We have included the reference to the recent Genetics paper by Mike Nonet.

Minor corrections/suggestions (not comprehensive, just things I noted):

Remove the extra comma after “simple” in p.3 line 10

We have modified the sentence.

p. 4” “Our first aim was to enable broadly deployable identification of PATCs.” I suggest: “We developed a user-friendly, versatile web server to identify and quantify PATCs.”

Agreed. We have exchanged for this paragraph.

p.7, first full paragraph “refractory” is a better term in this context than “recalcitrant”

Agreed. We have modified this paragraph and avoided using the term "recalcitrant".

Lines 25-26: “...as well as a fundamental lack of rules for how to best generate silencing-resistant transgenes.” Suggest: “...lack of clarity about best practices for creating silencing-resistant transgenes”

Agreed. We have rephrased to include this wording.

I’m not sure whether Figure 1 is useful – I’m not sure whether it would be printed/displayed in a way that the text would be readable, so a callout to the actual web interface may be more useful.

We have modified Figure 1 substantially. For readers that mainly browse the figures we believe that showing the website and web address is important.

“A comparison between a commonly used Peft-3:Cas9 plasmid (pDD133, (Dickinson et al., 2013)) and a similar PATC-rich Cas9 construct (Psmu-1:Cas9(PATC+)) showed higher insertion frequencies, in some cases more than double, for three researchers with different levels of experience (Figure 6C).” What is the conclusion of this analysis? It’s not clear how these plasmids were compared or which construct actually worked better.

Agreed. We have modified the panel and also clarified the comparison:

"A comparison between a commonly used Peft-3::Cas9 plasmid (pDD133) and Psmu-1::Cas9_{PATC} showed modest but significantly higher insertion frequency after optimization (Fig. 7c)."

I don't understand the following statements:

"We note that the requirement for PATCs internal to the transgenes itself is in line with the proposed biological role for PATCs; if insertion of foreign DNA into a PATC-rich genomic environment alone conferred resistance to silencing then it would be difficult to imagine how An/Tn sequences could participate in a genome defense."

What are "PATCs internal to the transgene itself..."? and what are the implications for the mechanism of PATC-mediated anti-silencing?

Agreed. We have restructured that paragraph and no longer speculate on the possible biological role of PATCs in genome defense mechanisms.

Reviewer #2 (Remarks to the Author):

In this manuscript by Aljohani et al, the authors set out to develop a method for transgene expression in *C. elegans* from extrachromosomal arrays that overcomes transcriptional silencing. Extrachromosomal arrays are a common method for expression of transgenes in somatic *C. elegans* tissues. However, arrays are silenced in the germline, which has limited their usefulness in this important tissue. Germline silencing is not unique to *C. elegans* and likely reflects mechanisms designed to protect the germline from invaders, such as transposons.

The authors show that the addition of Periodic An/Tn Clusters (PACTs) and removal of piRNA recognition sequences results in robust and stable transgene expression in the germline from extrachromosomal arrays. Moreover, addition of PACTs improves the efficiency of CRISPR-Cas9 edits and transposon-mediated single-copy insertions by improving genomic editing in the germline.

Overall the data is of high quality and the methods and tools developed here will be broadly useful to the *C. elegans* community. There is also potential for the information provided here to influence analysis of germline silencing in other species. I strongly support publication of this paper and list below a few minor issues that the authors should address in their revision.

We thank the reviewer for the kind comments.

Comments:

- Throughout the manuscript the authors use a specific nomenclature to refer to the different GFP versions that they used (standard GFP, ce-GFP and ce-GFP(PACT)). However, I was confused by some of the versions that they show in the supplement. For instance, what is wGFP? (Fig. S1C). In Figure S2B, was ce-GFP used? (it contains two syntons, whereas standard GFP has three). In Figure S4A, why does ce-GFP contain four syntons, while in the rest of the paper it only contains two?

Agreed. We have clarified the nomenclature throughout the manuscript and figures (including the supplementary figures). For the specific instances pointed out by the reviewer:

Figure S2B was in fact a codon-optimized version of *gfp* (*ce-gfp*) and has now been annotated as such (now Fig S3b).

Figure S4A. We have used several different version of codon-optimized *gfps*. In some cases, the *gfp* was optimized with two, three, four, or even five introns. The coding sequence was kept constant as much as possible except when restriction sites required minor changes.

wGFP was a reference to "worm GFP" which we have since changed to "ce-gfp".

We have included annotated DNA sequences for all constructs used in the study for further clarification.

- Another source of confusion was the data in figure S2B. The authors show robust expression of GFP from the *smu-1* promoter using a variety of 3'UTRs, which contradicts the results from Fig. 3C, which shows no expression of standard GFP under similar conditions (unless *ce-GFP* was used in fig. S2B, in which case they need to specify this).

Correct. The confusion was caused by our mislabeling of the *gfp* used in Figure S2B (it is a *ce-gfp*). There is, therefore, no discrepancy between Fig. 3C (now Fig 2B) and Figure S2B (now Fig S3b).

- In Figure 3B, why does GFP localize to the nuclei in the germline?

All GFPs contain a dual NLS (SV40 at N-terminus and *egl-13* at C-terminus) to facilitate visualization in germ cells. The figures are already somewhat "crowded" and we have therefore not indicated the tags visually. We have added the following sentence to the supplemental methods to clarify:

"All fluorescent proteins tested in isolation contain a dual NLS (SV40 at N-terminus and egl-13 at C-terminus) for easier germ cell detection."

- In Figures 5A and 5B, it is very hard to see the signals that the arrows are pointing to. It might be better to show these in grayscale instead of green. It might also help to zoom in to the germline region in Fig. 5A.

We believe this difficulty is caused by the conversion of the images to a compressed pdf file. We have attached a higher resolution image to the revised submission which has improved visibility on our computer monitor.

- In page 5, the authors say that their method for Golden Gate Cloning will be detailed in the Supplementary Protocol, but this is missing from the files provided.

Apologies. The Supplementary Protocol was not part of the submission. The protocol has been included in the Supplementary Methods (*Nature Communication* guidelines do not allow a separate Supplementary Protocol).

- The legend for Fig. S2 that is immediately below the figure is incorrect.

Correct. We have revised the Supplementary Figures substantially and inserted the correct figure legends.

- Figure S2A: Can the authors comment on why inserting GFP at the N-terminus results in silencing?

Yes.

We have now included these observations in the main text of the manuscript and Figure 2F. These results are consistent with observations in Shirayama et al. (2012). The authors show in Figure 1 that MosSCI insertions of *rde-3::gfp* and *cdk-1::gfp* are expressed at high frequency whereas *gfp::rde-3* and *gfp::cdk-1* are almost completely silenced. We believe that the N-termini of transgenes are more sensitive to silencing and we have expanded substantially on this aspect by testing a transgene's sensitivity to early splicing (Figure 3).

Reviewers' Comments:

Reviewer #1:

Remarks to the Author:

This revised manuscript is greatly improved from the original submission; I greatly appreciate the authors' diligent efforts to address the referees' comments and concerns. The addition of some new findings and removal of some of the more confusing components have strengthened the presentation of the work and its likely impact. The text still occasionally lacks clarity; ideally the reader would be able to understand the rationale and design of experiments and their interpretation from the text alone, which I did not always find to be the case, but the figures are very nicely presented and provide a great deal of useful information. The text will likely benefit from a careful copy editor.

I suggest a couple of minor semantic adjustments: "coding sequence" does not generally include introns or other noncoding information. "Transgene," in my view, describes a complete gene with its associated introns and upstream and downstream regulatory information, and not (for example) the insertion of a coding sequence (\pm introns) into an existing gene. A "fluorophore" is the fluorescent moiety at the core of a fluorescent protein - genes encode fluorescent proteins, rather than fluorophores.

Aside from these minor quibbles, I strongly support the publication of the work, which will be helpful the *C. elegans* research community and potentially interesting to other researchers interested in gene silencing mechanisms.

Reviewer #2:

Remarks to the Author:

In my initial review of the manuscript by Aljohani et al, I stated support for publication, while identifying minor issues that the authors needed to address. In their revised version, not only have the authors satisfactorily addressed all the initial concerns, but they have added experiments in which they carefully and systematically analyzed the requirements for PACT sequence insertions in preventing transgene silencing. The new analysis provides useful insights into the mechanisms that promote transgene silencing, a topic of broad interest. Moreover, the tools generated by this study will be of enormous help to the *C. elegans* community. I strongly support publication of the manuscript in its current form.

REVIEWERS' COMMENTS

Our replies in blue.

Reviewer #1 (Remarks to the Author):

This revised manuscript is greatly improved from the original submission; I greatly appreciate the authors' diligent efforts to address the referees' comments and concerns. The addition of some new findings and removal of some of the more confusing components have strengthened the presentation of the work and its likely impact. The text still occasionally lacks clarity; ideally the reader would be able to understand the rationale and design of experiments and their interpretation from the text alone, which I did not always find to be the case, but the figures are very nicely presented and provide a great deal of useful information. The text will likely benefit from a careful copy editor.

We thank the reviewer for the many constructive and valuable suggestions during the review process. We are glad that the reviewer shares our opinion that the added data strengthens the manuscript.

I suggest a couple of minor semantic adjustments: "coding sequence" does not generally include introns or other noncoding information. "Transgene," in my view, describes a complete gene with its associated introns and upstream and downstream regulatory information, and not (for example) the insertion of a coding sequence (\pm introns) into an existing gene. A "fluorophore" is the fluorescent moiety at the core of a fluorescent protein - genes encode fluorescent proteins, rather than fluorophores.

We have modified the relevant usage of "coding sequence", "transgene", and "fluorophore".

Aside from these minor quibbles, I strongly support the publication of the work, which will be helpful the *C. elegans* research community and potentially interesting to other researchers interested in gene silencing mechanisms.

Thank you.

Reviewer #2 (Remarks to the Author):

In my initial review of the manuscript by Aljohani et al, I stated support for publication, while identifying minor issues that the authors needed to address. In their revised version, not only have the authors satisfactorily addressed all the initial concerns, but they have added experiments in which they carefully and systematically analyzed the requirements for PACT sequence insertions in preventing transgene silencing. The new analysis provides useful insights into the mechanisms that promote transgene silencing, a topic of broad interest. Moreover, the tools generated by this study will be of

enormous help to the *C. elegans* community. I strongly support publication of the manuscript in its current form.

Thank you. We hope the rules and reagents will prove to be valuable to the *C. elegans* community.